# Retinotectal circuitry of larval zebrafish is adapted to detection and pursuit of prey

Dominique Förster[1†], Thomas O Helmbrecht[1,2†], Duncan S Mearns[1,2], Linda Jordan[1], Nouwar Mokayes[1], Herwig Baier[1*]

[1]Max Planck Institute of Neurobiology, Department Genes – Circuits – Behavior, Martinsried, Germany; [2]Graduate School of Systemic Neurosciences, LMU BioCenter, Martinsried, Germany

**Abstract** Retinal axon projections form a map of the visual environment in the tectum. A zebrafish larva typically detects a prey object in its peripheral visual field. As it turns and swims towards the prey, the stimulus enters the central, binocular area, and seemingly expands in size. By volumetric calcium imaging, we show that posterior tectal neurons, which serve to detect prey at a distance, tend to respond to small objects and intrinsically compute their direction of movement. Neurons in anterior tectum, where the prey image is represented shortly before the capture strike, are tuned to larger object sizes and are frequently not direction-selective, indicating that mainly interocular comparisons serve to compute an object's movement at close range. The tectal feature map originates from a linear combination of diverse, functionally specialized, lamina-specific, and topographically ordered retinal ganglion cell synaptic inputs. We conclude that local cell-type composition and connectivity across the tectum are adapted to the processing of location-dependent, behaviorally relevant object features.

**\*For correspondence:**
hbaier@neuro.mpg.de

[†]These authors contributed equally to this work

**Competing interests:** The authors declare that no competing interests exist.

## Introduction

Theories of efficient sensory coding (*Barlow, 1961*) often make the implicit assumption that the goal of sensory processing is a veridical representation of the external world. However, it is clear that the ultimate arbiter of efficiency is natural selection and that genetic information, developmental time, space, and material impose constraints on the design of the nervous system. Each of these evolutionary constraints has contributed to the neural implementations as we witness them in today's animal brains, making the ultimate goal of calculating an optimization function difficult to achieve (*Chalk et al., 2018*; *Dan et al., 1996*; *Machens et al., 2005*; *Simoncelli, 2003*). To understand why circuits are organized as they are and develop as they do, it is of paramount importance to identify constant and pervasive selective pressures that arise from the species-specific lifestyle of the animal. This study provides experimental support for the notion that the local statistics of the sensory environment, which changes dynamically as the animal interacts with the outside world, shape the topographic specializations of higher-order sensory and sensorimotor circuitry.

For many decades the retinotectal projection of zebrafish has served as a paradigmatic example for a visual map. Retinal inputs to the tectum are ordered retinotopically such that the position of an object in the visual field matches a corresponding focus of activity in tectal space (e.g., *Muto et al., 2013*). Neighborhood relationships in the environment, as they are projected onto the two-dimensional sheet of photoreceptors in the retina, are represented by neural activity in neighboring regions of the tectum. Visual stimuli in the front of the larva are detected by temporal regions of the retina, which transmit pre-processed information via the axons of retinal ganglion cells (RGCs) to anterior regions of the tectum. Similarly, stimuli in the peripheral visual field behind the animal activate nasal retina and posterior tectum, respectively (*Figure 1A*). The tectum then ultimately transforms visual information into behavioral commands (e.g., *Helmbrecht et al., 2018*).

**eLife digest** The retina is the thin layer of tissue in the eye that can receive light stimuli and convert them into electric signals to be transmitted to the brain. The cells that sense fine detail cluster at the center of the retina while the motion-sensing cells that keep track of movement lie at the periphery.

When zebrafish larvae hunt, their motion-sensing cells are triggered as a prey crosses their peripheral field of view. They then turn and swim towards it. As they approach, the prey image moves to the detail-sensing part of the retina and appears larger, filling more of the field of view at close range. The signals are then processed in defined parts of the brain, in particular in a region called the optic tectum. How this area is organized in response to the organization of the eye and the requirements of the hunt is still unclear.

Förster et al. set out to explore how the hunting routine of zebrafish larvae shapes the arrangement of neurons in the optic tectum. The larvae were exposed to different images representing the various aspects of the prey capture process: small moving dots represented passing prey at a distance, while large moving dots stood for prey just before capture. Measuring activity in the neurons of the optic tectum revealed that, like the eye, different areas specialize in different tasks. The back of the tectum was frequently activated by small dots and worked out which direction they were moving in during the first hunting steps. The front of the tectum responded best to large dots, often ignoring their direction, and helped the larvae to track their prey straight ahead. To test these findings, Förster et al. destroyed the large object-responsive cells with a laser and watched the larvae hunting real prey. Without the cells, the fish found it much harder to track and catch their targets.

These results shed light on the link between behavior and how neurons are arranged in the brain. Future work could explore how the different neurons in the optic tectum are connected, and the behaviors they trigger in the fish. This could help to reveal general principles about how sensory information guides behavior and how evolution has shaped the layout of the brain.

The neuropil of the larval zebrafish tectum is spatially organized along the superficial-to-deep axis into layers, ten of which are receiving input from dedicated subsets of RGCs (*Robles et al., 2013*; *Robles et al., 2014*). The remaining layers are innervated by axons from the somatosensory lateral line (*Thompson et al., 2016*) or contain dendrites and axons of interneurons and projection neurons (*Helmbrecht et al., 2018*). The tectal neuropil layers are schematically depicted in *Figure 1B*. Recent work has revealed an enormous functional and morphological diversity of RGC types, which serve as local feature detectors for specific aspects of the visual scene, such as direction of motion, onset or offset of light, object size or chromaticity. Earlier studies have shown that individual RGCs select one layer each, in which they arborize and make synapses onto tectal dendrites (*Xiao and Baier, 2007*). Thus, each retinorecipient layer contains a complete, yet feature-selective, map of visual space. RGCs that respond to visual features resembling the speed and size of prey project to the most superficial layer (SO; *Semmelhack et al., 2014*; see *Figure 1B*), whereas RGCs that are specifically tuned to a rapidly expanding (looming) dark object, simulating an approaching predator or an obstacle on a collision course, terminate in deeper layers (SFGS5/6; *Temizer et al., 2015*; see *Figure 1B*).

Asymmetries in visual feature processing have been recognized across the retina of several vertebrates (for a recent review, see *Baden et al., 2020*). Prime examples for such functional specializations are the fovea of primates (*Sinha et al., 2017*), the asymmetric distributions of RGC types and photoreceptors in mice (*Baden et al., 2016*; *Bleckert et al., 2014*; *Szatko et al., 2020*; *Warwick et al., 2018*) and of bipolar cells, photoreceptors and RGCs in zebrafish (*Yoshimatsu et al., 2020*; *Zhou et al., 2020*; *Zimmermann et al., 2018*). The two retinotopic dimensions of the tectum, the anterior-posterior and the dorsal-ventral axis, have so far received little attention in this regard. Zebrafish larvae do not possess a *prima facie* fovea, although they have evolved a high-acuity subarea in the temporal-ventral quadrant of the retina in which RGCs are more densely packed than in the periphery (*Schmitt and Dowling, 1999*; *Zhou et al., 2020*). This region holds the image of prey in the final phase of hunting behavior and, similar to the mammalian fovea,

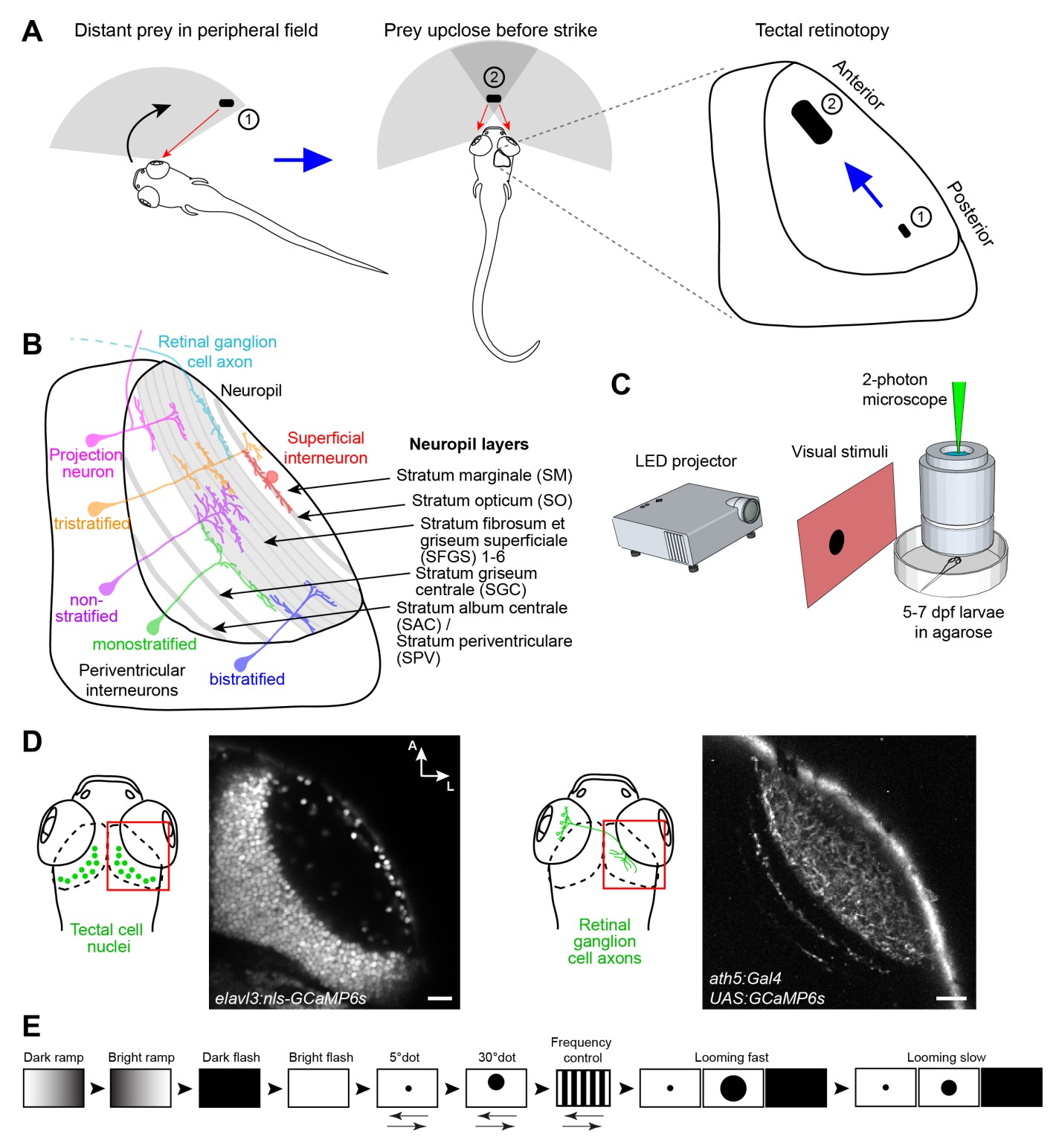

**Figure 1.** Experimental paradigm for studying location-specific processing in the tectum. (**A**) In a typical hunting sequence, the fish detects prey in its peripheral visual field (1), ultimately turns and approaches to bring the prey image into its central binocular field (2). Hypothetically, the retinotectal map might be adapted to this location- and size-specific representation of the prey object. (**B**) Sketch of the tectum showing previously described cell types and neuropil layers. (**C**) Schematic for functional imaging setup. (**D**) On the left: Region of interest (ROI) for imaging tectal cell responses and exemplary expression of nuclear-localized GCaMP6s. On the right: ROI for RGC imaging and expression of GCaMP6s in RGC axons under control of *ath5:Gal4*. (**E**)

*Figure 1 continued on next page*

*Figure 1 continued*

Stimulus protocol. Arrows below stimulus representation indicate object movement, first in nasal, then in temporal direction. See Materials and methods for details. Scale bar in (D): 20 µm.

occupies a disproportionately large area of the visual map in the tectum. Despite a wealth of data on tectal neuron morphologies (see *Figure 1B*; *Förster et al., 2017*; *Nevin et al., 2010*; *Robles et al., 2011*; *Scott and Baier, 2009*), systematic changes in cell-type composition or connectivity along the anterior-posterior or dorsal-ventral axes of the tectum, resulting in gradients or other asymmetries of feature selectivity, have just begun to be revealed (*Wang et al., 2020*).

Here, we ask if such asymmetries can be predicted from first principles and related to the behavioral ecology of the zebrafish larva. As the animal interacts with a visual object through its own movements, relevant stimulus features continually change within the retinotopic coordinate frame. For example, in a typical hunting cycle, a zebrafish larva detects a prey item at a distance in its peripheral, monocular visual field (*Mearns et al., 2020*; *Patterson et al., 2013*). Posterior tectal circuits might therefore have evolved to respond to small-sized objects of ca. 5° and to locally compute their direction of movement. As the fish turns toward and approaches the prey, the stimulus enters the central, binocular visual field and expands to ca. 30° in visual angle (*Figure 1A*). Activation of the anterior tectum has previously been described during this late hunting phase (*Muto et al., 2013*). Neurons in the anterior tectum should therefore be tuned to larger object sizes and may rely on interocular comparisons to compute the object's displacement from the midline. At all positions, the tectum should be able to distinguish between prey and looming threats and process them separately (*Barker and Baier, 2015*). The laminar segregation of functional channels, which are established by RGC inputs, is therefore expected to be maintained by tectal circuits independent of retinotopic location.

Using volumetric two-photon calcium imaging to map out the feature space along the anterior-posterior retinotopic axis and across the layers of the tectum, we discovered a neural substrate for each of above predictions. Moreover, we show that the broad range of tectal responses originate to a large extent, but not exclusively, from a linear combination of functionally diverse RGC inputs. The dendrites of tectal cells are positioned in layers that predict their stimulus selectivity. We conclude that the cellular architecture underlying local processing in the tectum is adapted to the expected features of a prey object as it moves across the visual field during a hunting pursuit.

## Results

### Tectal neurons respond to a broad range of visual features

To broadly sample responses to object features, we designed a battery of simplified visual stimuli and controls. We employed two-photon calcium imaging of 5 to 7 dpf old larvae, which received monocular visual stimulation (*Figure 1C*). At this larval stage, panneuronal expression of the nuclear-localized calcium indicator GCaMP6s (driven by the *elavl3* promoter) labels on average 5793 ± 202 cells per tectum (n = 10 fish; mean ± SEM) (*Figure 1D*). The stimulus set consisted of a moving dot of 5° ('small'), which approximates the size of prey at the onset of hunting behavior (*Bianco and Engert, 2015*; *Patterson et al., 2013*; *Semmelhack et al., 2014*), a moving dot of 30° ("large"), which is the approximate size of prey directly before the capture strike, and an expanding disc at different velocities, which simulates an approaching object and is able to evoke escape responses (*Bhattacharyya et al., 2017*; *Dunn et al., 2016*; *Temizer et al., 2015*). We further added controls for global luminance changes (dark and bright ramps and flashes), as well moving gratings with high spatial (5°) and temporal frequency as a negative control for small-dot responses (*Figure 1E*; see Materials and methods). With this battery of visual stimuli, we obtained reproducible calcium responses in up to 30% of all tectal cells per imaging plane. We created 15 regressors for the different stimulus variants and calculated a score value for each tectal cell (*Figure 2A*). To classify functional response types, we performed hierarchical clustering of representative response vectors obtained by affinity propagation (see Materials and methods). This resulted in a dendrogram for 76 exemplars, which are representative of the 1759 sampled tectal cells in total (*Figure 2B–D*, and *Figure 2—figure supplement 1A*). A silhouette analysis to validate the clustering showed that a

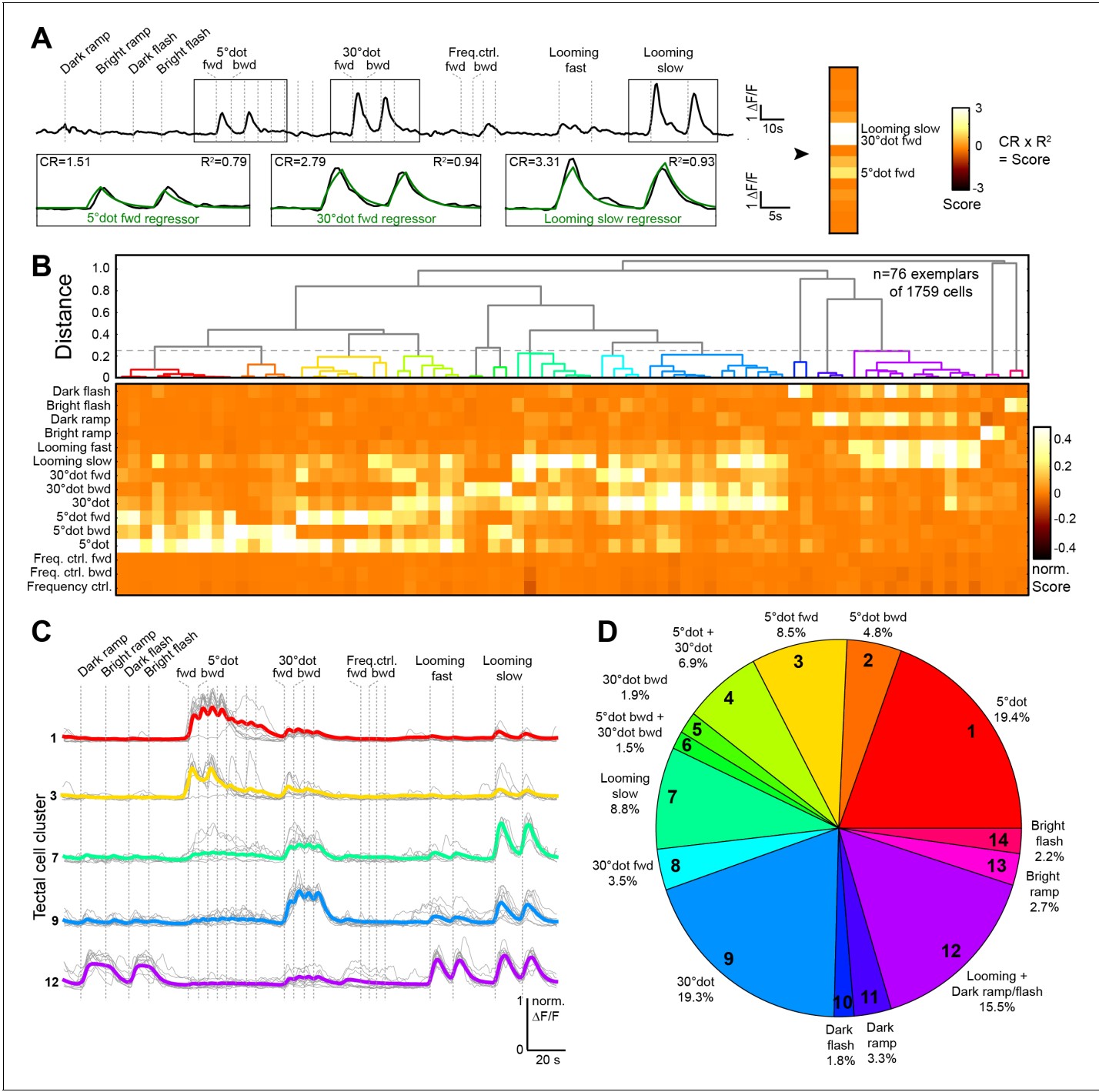

**Figure 2.** Behaviorally relevant response clusters in the tectum. (**A**) Analysis of calcium imaging data. Within selected response windows (black rectangles), the ΔF/F traces were correlated to the corresponding regressor and 15 score values were calculated for each cell (CR: coefficient of regression, $R^2$: correlation, response: black trace, model: green trace). (**B**) Hierarchical clustering of functional cell types in the tectum. Normalized scores for 76 exemplars, obtained by affinity propagation of 1759 cells (of three larvae) in total are shown. Dashed line indicates a chosen distance threshold of 0.25, which results in 14 functional clusters. (**C**) Normalized calcium transients of all exemplars (gray) and average traces of all cells (colored) for the five largest clusters. (**D**) Functional cluster distribution. Tectal cluster numbers are indicated.

The online version of this article includes the following figure supplement(s) for figure 2:

**Figure supplement 1.** Functional clustering of tectal cells.

minimal number of 14 clusters yielded an optimal classification of the data (*Figure 2—figure supplement 1B*; see Materials and methods).

To investigate the dimensional structure of the different response profiles, we performed principal component analysis on the scores for all tectal cells. Plotting the three main principal components (PCs), which could explain 74.9% of the variance in the dataset, aligned the scores along three axes for small-dot, large-dot, and looming/luminance- (OFF-) responding cells (*Figure 2—figure supplement 1C and D*). To show that the measured tectal cell responses were significantly different from chance, we shuffled the scores for each regressor 1000 times and calculated the PCs. Taking the average of the explained variance per shuffling, we consistently found a lower average explained variance, that is 57.6% for the three main PCs (*Figure 2—figure supplement 1D*), indicating that tectal cells do not respond randomly to our set of stimuli.

## Responses of tectal neurons are enriched for various forms of object motion

Overall, we found a broad spectrum of different response types in the tectum. Few cells responded to only one of the presented stimuli; most cells we imaged were multi-responsive (*Figure 2B*). 43.6% of all cells responded to a looming stimulus (with a score >0.2), 41.1% responded to a small dot, and 33.1% responded to a large dot (*Figure 2D*). Only a small number of cells responded to a bright ramp (2.7%) or a bright flash (2.2%), and these cells were rarely sensitive to other stimuli. Responses to dark ramp and dark flash often coincided with each other and with responses to looming stimuli (fast and slow), but rarely overlapped with responses to small or large moving dots. Responses to a slow-looming stimulus showed a gradual overlap with moving-dot responses; more than half of all cells that were sensitive to a large dot also responded to a slow-looming stimulus. The 5° grating did not trigger significant responses in the tectum, suggesting a selectivity to individual objects rather than to high spatial frequency.

Next, we characterized the tuning properties of tectal cells whose somata reside inside the tectal neuropil. Superficial interneurons (SINs), with cell bodies in the SO to SFGS1 neuropil layers, have previously been reported to receive size-tuned retinal inputs (*Del Bene et al., 2010*; *Preuss et al., 2014*). The largest fraction of SINs was mapped to our large-dot responsive cluster (~45%; *Figure 2—figure supplement 1E*), whereas only a small number of SINs (~6%) were sensitive to a 5° dot. Neuropil interneurons (NINs), residing within deeper layers of the neuropil, predominantly belong to the looming/dark ramp-responsive cluster (~30%), with about 20% of NINs responding to large dots (*Figure 2—figure supplement 1E*).

Taken together, the majority of tectal cells, both in the periventricular layer and embedded in the neuropil, respond to object motion, that is small, or large, or looming dots, sometimes in combination. A substantial fraction of cells responds to global dimming or looming (OFF cells). Very few cells respond to global brightening (ON cells). OFF and ON cells are largely non-overlapping with object-detecting cells.

## Tectal responses originate from diverse, feature-specific RGC inputs

We next asked to what extent the feature selectivity of tectal neurons is inherited from retinal inputs. In our imaging setup, we applied the same battery of visual stimuli to larvae expressing cytoplasmic GCaMP6s in RGCs (*Figure 1D*). A pixel-wise regressor and cluster analysis resulted in a dendrogram for 1157 exemplars, which were grouped into ten functional clusters (although four RGC clusters resulted in the highest silhouette coefficient, we chose 10 clusters, for a significantly higher modeling correlation score, as shown below) (*Figure 3A–C*, and *Figure 3—figure supplement 1*; see Materials and methods).

Overall, RGC responses were similar to tectal responses, but less specialized, with only few pixels responding exclusively to a single stimulus. Two thirds (67.2%) of the pixels responded to a large dot with a score greater than 0.2 (*Figure 3A* and *Figure 3—figure supplement 1A*). Generalized OFF responses to a dark ramp and a looming stimulus were similarly prominent. Non-intuitively, ON responses were sometimes combined with dark looming stimuli (RGC cluster no. 2 and 9; *Figure 3A* and *Figure 3—figure supplement 1A*), a tuning profile we did not observe in tectal cells. Interestingly, direction-selective responses to forward- (nasalward-) moving stimuli, especially to a large dot, were more abundant than for the opposite direction (*Figure 3A*). These units are expected to be

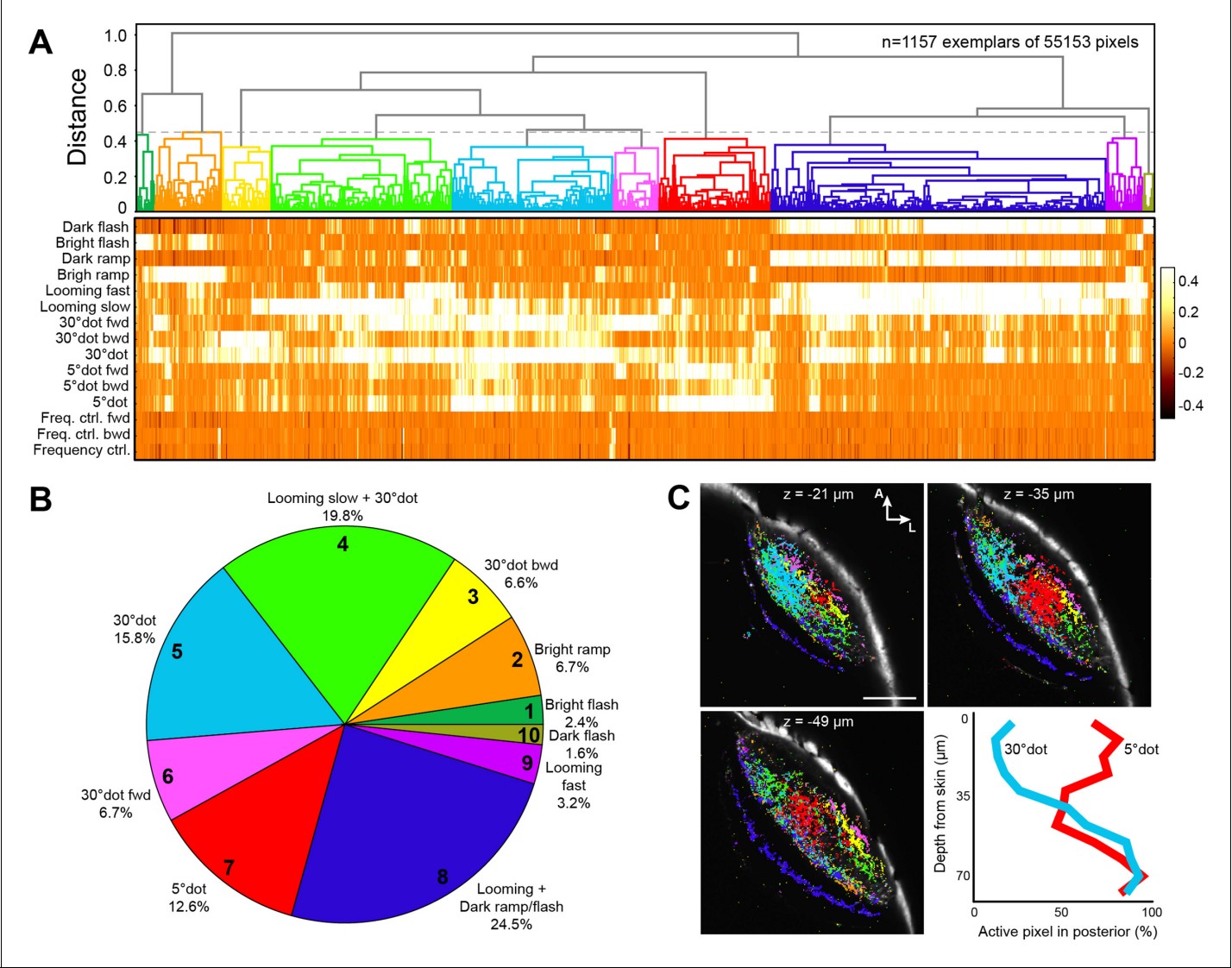

**Figure 3.** Clustering of functional RGC responses in the tectum. (**A**) Hierarchical clustering of functional RGC pixels. Normalized scores for 1157 exemplars, obtained by affinity propagation of 55,153 pixels in total are shown. Dashed line indicates a chosen distance threshold of 0.45, which results in 10 functional clusters. (**B**) Functional cluster distribution of all analyzed RGC pixels. Cluster numbers are indicated. (**C**) Spatial distribution of functional RGC pixels in the tectal neuropil. Pixels were cluster-color-coded and overlaid onto single planes of the *ath5:Gal4 UAS:GCaMP6s* expression pattern. Z indicates plane position as the distance from dorsal skin (z = 0 µm). Last panel in (**C**) shows quantification of 30˚ dot-responsive (blue) and 5˚ dot-responsive (red) pixels in the posterior tectum along different z-planes. Scale bar: 50 µm.

The online version of this article includes the following figure supplement(s) for figure 3:

**Figure supplement 1.** Functional clustering of RGC types.

**Figure supplement 2.** Figure panels showing the active RGC pixels of three imaging planes from **Figure 3C** separately for the three relevant clusters (30˚ dot, 5˚ dot, looming+dark ramp/flash).

activated when an object approaches from behind or when the fish turns toward an object in its peripheral visual field.

## Tectal neurons linearly combine retinal inputs, but also compute de novo additional features

We asked to what extent we could quantitatively explain the sampled tectal responses by using ganglion cell input. This analysis can distinguish between two extreme scenarios: The tectum may either be a passive relay station for RGC inputs. Or, alternatively, it may 're-compute' the image based on

unrelated RGC inputs. We used a simple feed-forward, linear modeling approach (L1-regularized, Lasso) with non-negative constraints to predict tectal cell responses by a sum of weighted RGC inputs (*Figure 4A*; see Materials and methods). Modeling the scores for each of the 1759 tectal cells resulted in a high prediction quality (median correlation $R^2_{score}$ = 0.68, median RMSE = 0.06; *Figure 4B* and *Figure 4—figure supplement 1*). Similarly, we modeled the calcium transients for all tectal cells and calculated the correlation $R_{trace}$ between measured and predicted values (*Figure 4C and D*). We also tested how a varying score threshold for the RGC responses, and thus a different number of RGC clusters would change the modeling prediction quality. We found that the best prediction of tectal calcium transients ($R_{trace}$) can already be achieved by linear modeling of only four RGC clusters. Correlation for the tectal score values ($R^2_{score}$), however, increases significantly with ten RGC clusters (*Figure 4—figure supplement 1*).

Most tectal cell responses could be well explained by a linear combination of on average two RGC input clusters (*Figure 4D*);~36% of all responses could even be predicted by a single RGC input weight. However, specific tectal response features were modeled poorly: First, nearly all modeled tectal calcium traces showed responses to a large dot, owing to the high abundance of RGC responses to this stimulus (*Figure 4D*). Second, the weak RGC responses to a moving small dot resulted in a poor prediction of the DS tectal clusters no. 2 and 6 (*Figure 4D*). Third, modeling tectal calcium responses that are exclusive to ON or OFF stimuli was generally imperfect, and the worst correlation $R^2_{score}$ was found for the tectal gradual OFF-selective cluster no. 11. Our modeling results suggest that most visual representations in the tectum are directly inherited from RGC inputs. In addition, non-retinal, presumably intratectal computations add feature specificities, such as information on the direction of small moving objects, and sharpen both object-size and luminance selectivities of tectal neurons.

## Tectal layers process different object features according to their retinal inputs

We asked if tectal layers are distinct with respect to their feature selectivity. Along the superficial-to-deep axis, in line with previous publications, we found that RGC axons sensitive to small dots enter the tectum in superficial layers (SO to SFGS4) with a peak in SFGS1/2 (*Figure 3—figure supplement 1D*; *Preuss et al., 2014*). DS pixels were located most superficially in the posterior half of SFGS1 (*Figure 3C*; *Nikolaou et al., 2012*). OFF-responsive axons, on the other hand, arborized in deep SFGS layers, SGC and SAC/SPV, and most extensively in SFGS5/6 (*Figure 3—figure supplement 1D*; *Temizer et al., 2015*). To investigate if the dendrite morphologies of functionally identified tectal neurons matched these input layers, we carried out function-guided inducible morphological analysis (FuGIMA) of single tectal neurons (*Förster et al., 2018*). We used nuclear-localized GCaMP6f (nls-GCaMP6f) and regressor-based analysis to identify tectal cells that belong to the three largest clusters: small-dot responsive, large-dot responsive, and OFF cells. Co-expressed photoactivatable GFP (paGFP) was then used to fluorescently label a cell of interest with a two-photon laser pulse directed at the soma (*Figure 5A and B*). After allowing some time for diffusion of the activated GFP into the neurites, single cells were traced and registered to a standard brain together with RGC reference markers. This allowed us to quantify the extent of neurite arborization in each layer of the tectum (*Figure 5C–E*).

We compared our FuGIMA dataset (n = 91 cells) to a random collection of single tectal cells (n = 188; *Figure 5—figure supplement 1*), which were stochastically labeled with the BGUG method (*Xiao and Baier, 2007*). This analysis revealed that the three functional classes sampled branched preferentially in SFGS5/6. In addition, we found that small- and large-dot responsive cells showed significantly denser arborizations in SO, SFGS1/2, and SFGS3/4 compared to OFF cells. OFF cells, on the other hand, were biased to extend neurites in the SGC, the SAC, SAC/SPV and the SM (*Figure 5E*). SM is a layer at the surface of the tectum, which is innervated by the torus longitudinalis, a higher-order visual area with strong OFF responses (*Northmore, 1984*; *Robles et al., 2020*). SGC is a neuropil area abutting SFGS, in which multisensory information is processed. SAC is close to RGC axons that terminate in SAC/SPV and carry ambient luminance information to the tectum (*Kölsch et al., 2020*). A comprehensive catalog of all identified tectal interneuron morphotypes is shown in *Figure 5—figure supplement 2*.

We further investigated the extent of tectal cell arborizations by measuring the arbor areas in each layer (*Figure 5—figure supplement 3A*). We found that single cell arbors were generally small

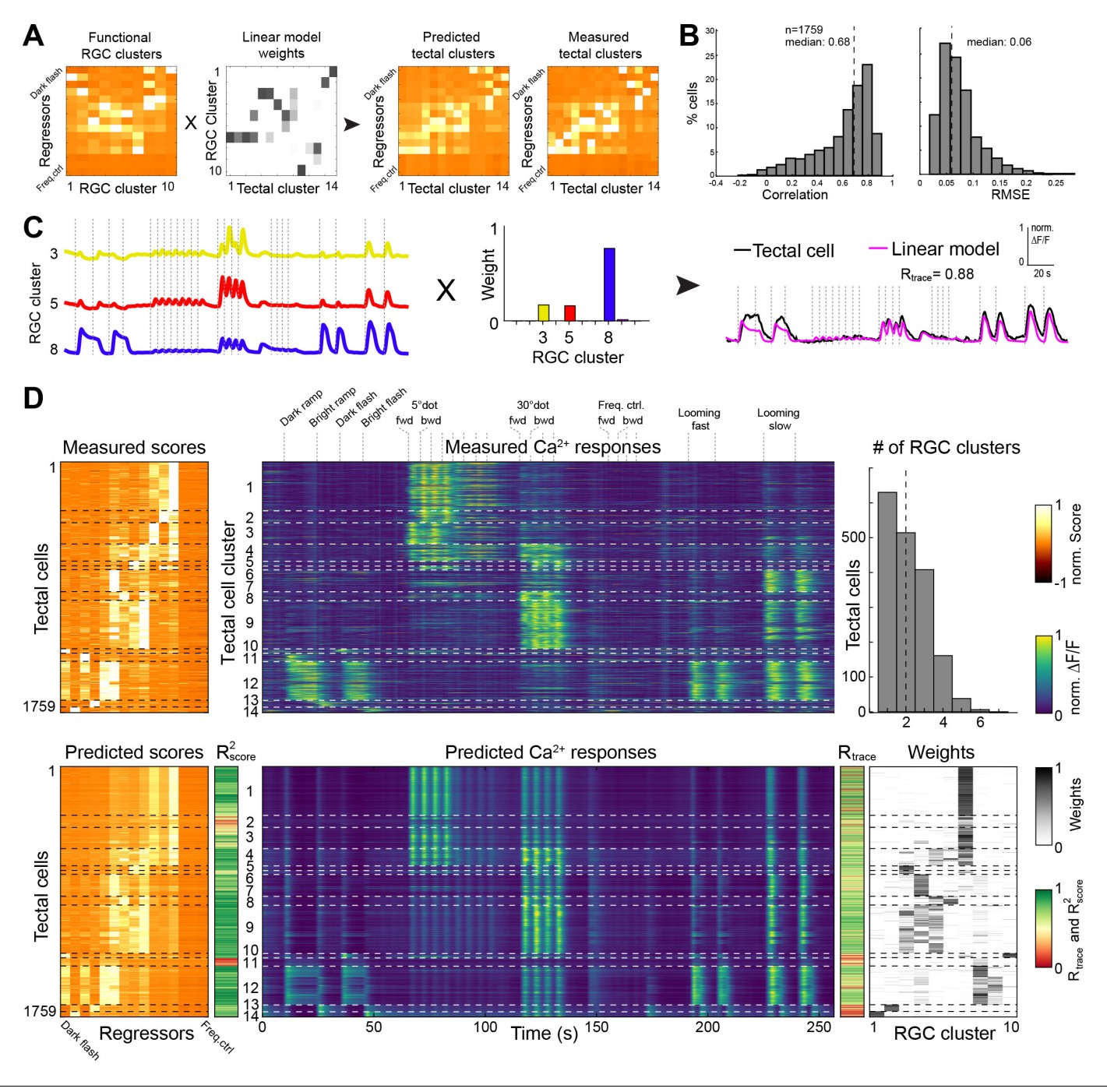

**Figure 4.** Modeling of tectal responses by linear combinations of RGC inputs. (**A**) Modeling workflow. Tectal cluster scores were predicted by a linear combination of weighted RGC cluster scores and finally compared to previously measured tectal scores. For color scale, see (**D**). (**B**) Prediction quality for modeling the scores of each sampled tectal cell (n = 1759). Left graph shows the correlation between predicted and measured scores. Right graph shows distribution of root mean squared errors of the cross-validated model (see Materials and methods for details). (**C**) Example for modeling the calcium response of a single tectal cell from weighted average responses of three RGC clusters. (**D**) Summary of modeling scores (left), calcium responses (middle), and weights (lower right) for all tectal cells (n = 1759). Functional tectal clusters are indicated by dashed horizontal lines. Color scales are shown on the right. Upper graph on the right shows the distribution of the number of RGC clusters used for modeling tectal responses. Dashed vertical line indicates a median of two RGC clusters.

The online version of this article includes the following figure supplement(s) for figure 4:

**Figure supplement 1.** Linear modeling parameters.

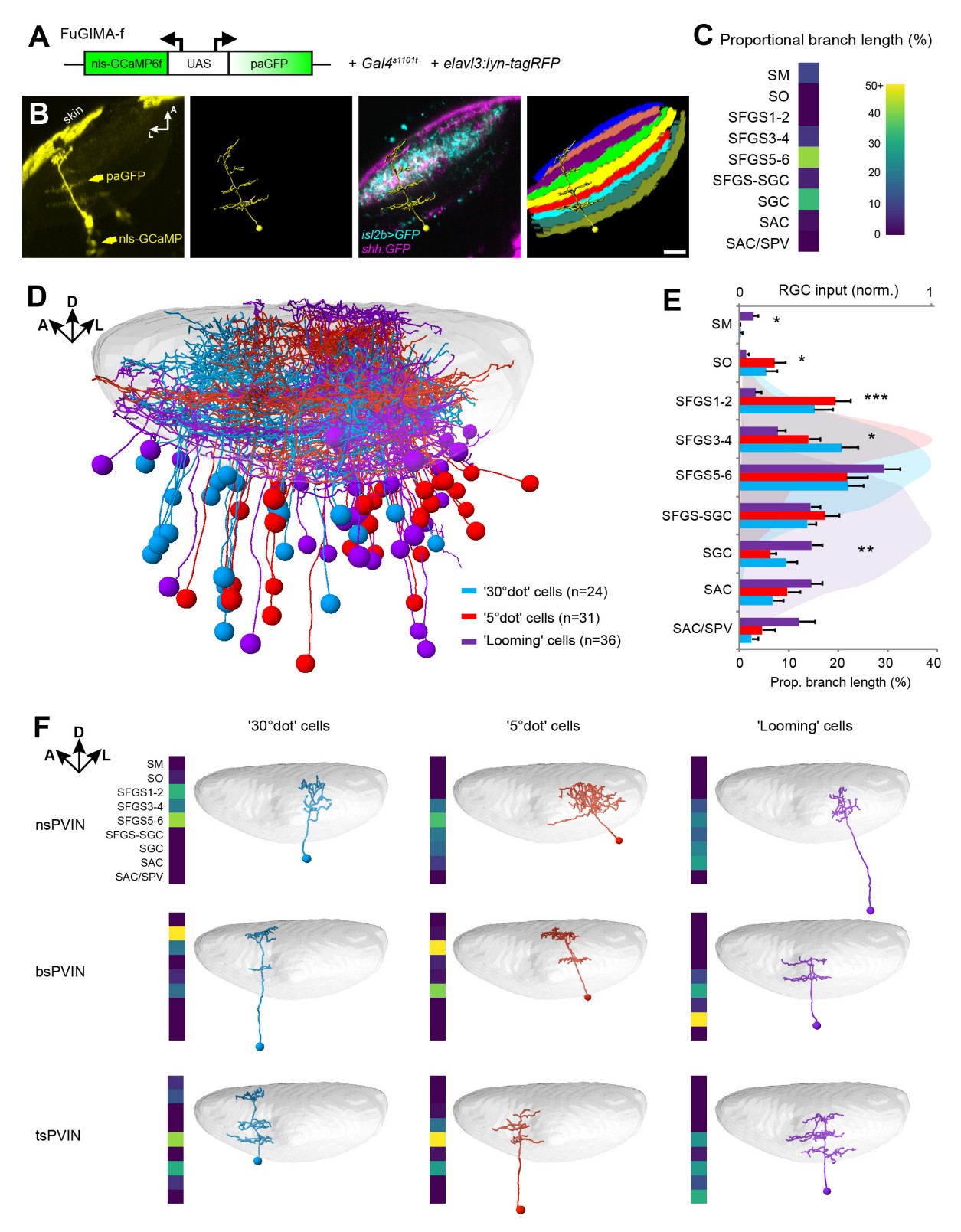

**Figure 5.** Dendrite morphologies of functionally identified tectal neurons match input layers. (**A**) The FuGIMA-f construct, which allows coexpression of nuclear-localized GCaMP6f and photoactivatable GFP (paGFP), was combined with *Gal4s1101t* for panneuronal expression and *elavl3:lyn-tagRFP* for image registrations. (**B**) Workflow of single-cell photoactivation, cell tracing, landmark registrations and layer quantifications (see Materials and methods for details). (**C**) Morphological barcode for the cell in (**B**). (**D**) Sideview of registered FuGIMA cells in the tectum of a standard brain. Tectal neuropil is

*Figure 5 continued on next page*

*Figure 5 continued*

shaded in gray. (**E**) Average proportional branch length of neurites in the respective tectal layers, quantified for 30° dot- (blue), 5° dot- (red), and looming- (purple) responsive cells. Statistically significant differences between 5° dot- and looming-responsive cells are indicated by stars. For comparison, the quantification of RGC input in the respective layers is shown in the back (see *Figure 3—figure supplement 1D*). Error bars are SEM. ***: p < 0.001, **: p < 0.01, and *: p < 0.05. (**F**) Exemplary tectal cell morphotypes identified for the response groups described above. PVIN: periventricular interneuron; ns: non-stratified; bs: bistratified; ts: tristratified. Scale bar in (**B**): 20 μm.

The online version of this article includes the following figure supplement(s) for figure 5:

**Figure supplement 1.** Comparison and quantification of tectal cell morphologies.
**Figure supplement 2.** Tectal interneuron catalog.
**Figure supplement 3.** Quantification of tectal cell arbor size.

in superficial layers (SM to SFGS3-4) and largest in deeper layers (SFGS5-6 to SAC/SPV; *Figure 5—figure supplement 3B*). When comparing the ratio of deep vs. superficial arbor size of multi-stratified cells, we found morphological differences between object-motion responsive and OFF cells. While on average, small-dot responsive cells have a columnar shape, OFF cells have extended arbors in deeper layers, rendering them cone-shaped (*Figure 5—figure supplement 3C and D*). We did not detect a systematic morphological difference between small- and large-dot responsive cells (*Figure 5F*). Object-motion responsive and OFF cells thus target layers that match their corresponding retinal and, in the case of SM, non-retinal inputs and also differ in more subtle morphological features.

## Retinotectal circuits are differentially tuned for object size and direction along the anterior-posterior axis

Along the anterior-posterior (A-P) axis, we found a separation of size-selective RGC terminals. RGC axons responding to a large dot were mainly located in the anterior-dorsal quadrant of the tectal neuropil, whereas small-dot responsive pixels were found in the medial to posterior part (*Figure 3C*). This compartmentalization is inherited by the corresponding tectal populations (*Figure 6A*). Compared to all sampled cells, the large-dot response cluster was shifted to the anterior tectum, while cell bodies responding to small dots were biased to the posterior region. The strongest posterior bias was found for direction-selective cell bodies, responsive to a small, forward moving dot (*Figure 6A*). We extended this analysis to our FuGIMA dataset, to quantify the extent of neurite arborizations in the neuropil. We found the same effect, that is large-dot responsive cells arborize more extensively in the anterior neuropil, compared to all sampled interneurons, while ~90% of neurites from DS small-dot responsive cells were found in the posterior half (*Figure 6B and C*). These findings indicate a spatial gradient of sensitivity to object size, which is introduced by the topographic order of RGC inputs and inherited by the retinotopic array of tectal cells.

## Ablation of small size-tuned RGC inputs removes tectal responses to small objects

To directly demonstrate that RGCs impose their feature selectivity onto postsynaptic tectal cells, we carried out an ablation experiment. From a previous study, we knew that small-dot responsive RGCs project specifically into SO after forming a collateral arbor in AF7, the neuropil of the parvocellular superficial pretectal nucleus (*Semmelhack et al., 2014*). By laser ablation of the RGC axon bundle that leaves AF7, we achieved selective disruption of small-object input to the SO layer (*Figure 7A–C*). Functional calcium imaging before and after the ablations revealed that small-dot responses were significantly diminished in tectal cells (*Figure 7D, E and G*). In contrast, the number of looming-responsive cells in the affected tectum was not reduced, but even increased in some animals, possibly due to the loss of inhibition by the small-object-processing circuit (see *Barker and Baier, 2015*; *Figure 7D, F and G*). These results indicate that RGC projections to SO are essential for tectal cells to assume their tuning to small-object motion.

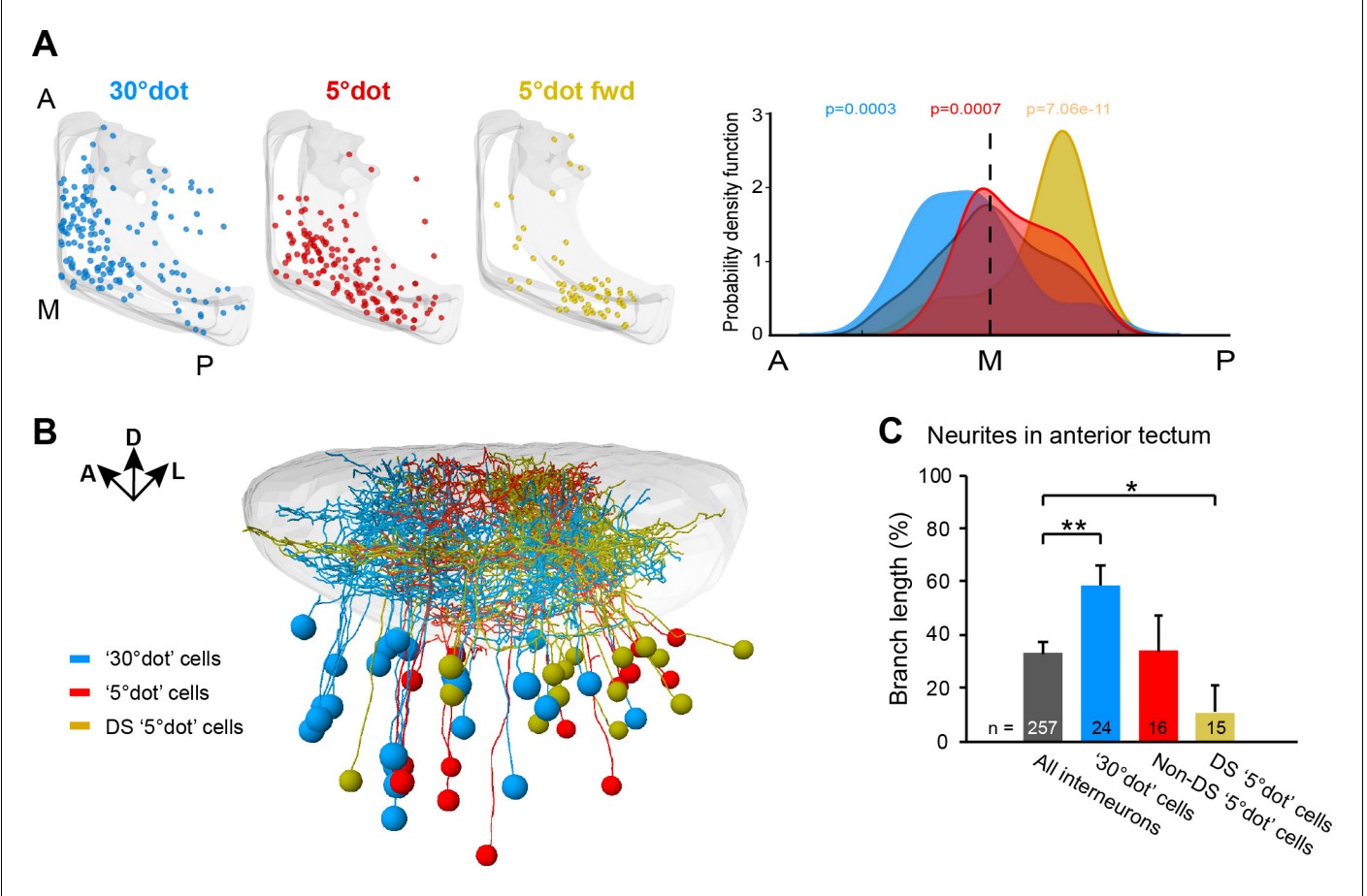

**Figure 6.** Functional compartmentalization of the tectum along the anterior-posterior axis. (**A**) Distribution of tectal cell bodies from 30°-dot (blue), 5°-dot (red) and 5°-dot-forward (yellow) response clusters. Anterior (A), medial (M) and posterior (P) positions of the tectum are indicated. Graph shows probability density function for cell body distribution. Integrals are colored according to their functional cluster with p-values characterizing the difference from the distribution of all sampled cells (gray integral). (**B**) Tectal sideview of registered FuGIMA neurons showing the distribution of 30°-dot (blue), 5°-dot non-DS (red) and 5°-dot-DS (yellow) cells. (**C**) Quantification of proportional neurite branch length of tectal cells in the anterior tectum. N equals number of cells. **: p = 0.006,; *:p = 0.014.

## Tectal representation of large (close) objects in frontal visual field is required for hunting

As the fish larva approaches a prey item, such as a paramecium or a rotifer, object size on the retina increases in visual angle. During hunting, the eyes converge and create an area of binocular overlap in the temporal retina. Convergent eye movements are accompanied by specialized turns, known as J-turns, that serve to center the prey in the visual field. Converged eyes and J-turns are characteristic of hunting episodes. We hypothesized that the large-dot responsive cells in the anterior tectum might be relevant for tracking prey at close range. To test this, we ablated between 3 and 15 single cells, which had been classified as large-dot responsive, in the right tectum (*Figure 8A*). Prey capture behavior was then analyzed in free-swimming larvae (*Mearns et al., 2020*).

Following removal of large-dot responsive cells, animals spent less time with their eyes converged, indicating less time spent engaged in hunting behavior (*Figure 8B*). In addition, their J-turns were biased to the right side, indicating defective prey detection by the left eye or right (ablated) tectum, respectively (*Figure 8C and D*). Control fish, in which entirely non-responsive cells were ablated, showed no effect on prey capture behavior and were indistinguishable from untreated or agarose-embedded larvae (*Figure 8A–D* and *Figure 8—figure supplement 1A–C*). Likewise, ablation of small-dot responsive cells, either in the anterior or posterior tectum did not result in

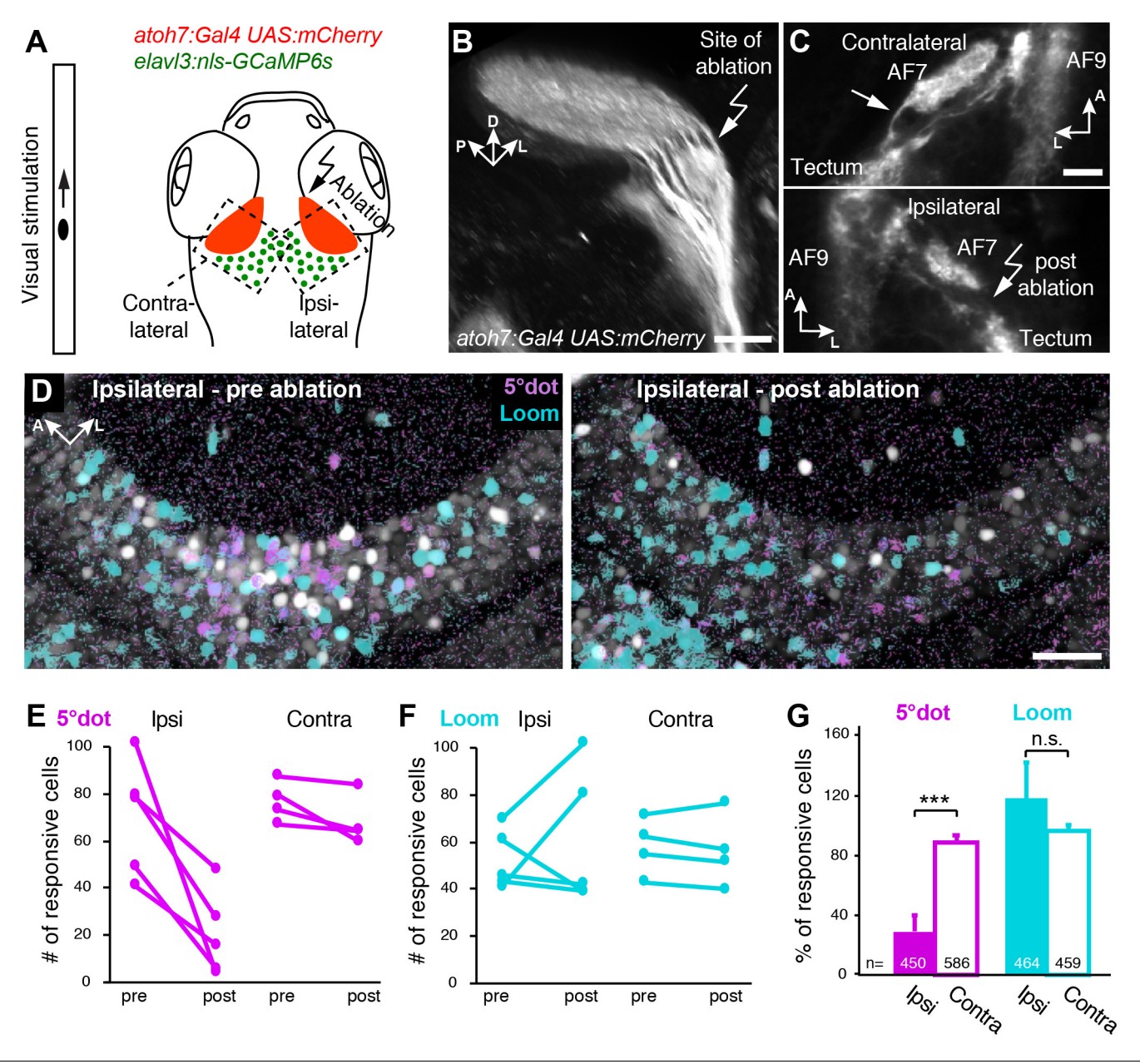

**Figure 7.** Small size-tuned RGC inputs are essential for small-object processing in the tectum. (**A**) Experimental setup for RGC axon ablations. Larvae are expressing mCherry in RGCs and nulear GCaMP6s panneuronally. The eye contralateral to the ablation site is visually stimulated and the ipsilateral tectal cells are functionally imaged before and after the ablations. As a control, the eye ipsilateral to the ablation site is stimulated and the contralateral tectal cells are imaged in the same fish. (**B**) Sideview of mCherry expression in RGCs at 6 dpf shows the most lateral axon bundle, which leaves AF7 for the SO layer (arrow). (**C**) Dorsal view of single image planes showing the axon fibers of interest in the contralateral (control, upper panel) and ipsilateral (ablated, lower panel) pretectum of the same fish. (**D**) Single functional image planes, projected over time, showing nuclear GCaMP6s expression in the ipsilateral tectum, before (6 dpf, left) and after (7 dpf, right) ablation. Pixels are color-coded by preference for 5˚ dot (magenta) or looming (cyan) stimuli. (**E**) Number of cells per image plane (out of two fish), which are responsive to a 5˚ dot stimulus, before and after ablations in the ipsilateral and the contralateral tectum. (**F**) Same as (**E**), showing the number of cells responsive to a looming stimulus. (**G**) Fraction of 5˚-dot- and looming-responsive cells after ablations in the ipsilateral and contralateral tectum. Error bars are SEM. ***: p = 0.0006; n.s.: p = 0.46. N equals number of cells from two independent fish. Scale bars in (**B**): 30 μm, (**C**): 20 μm, and (**D**): 50 μm.

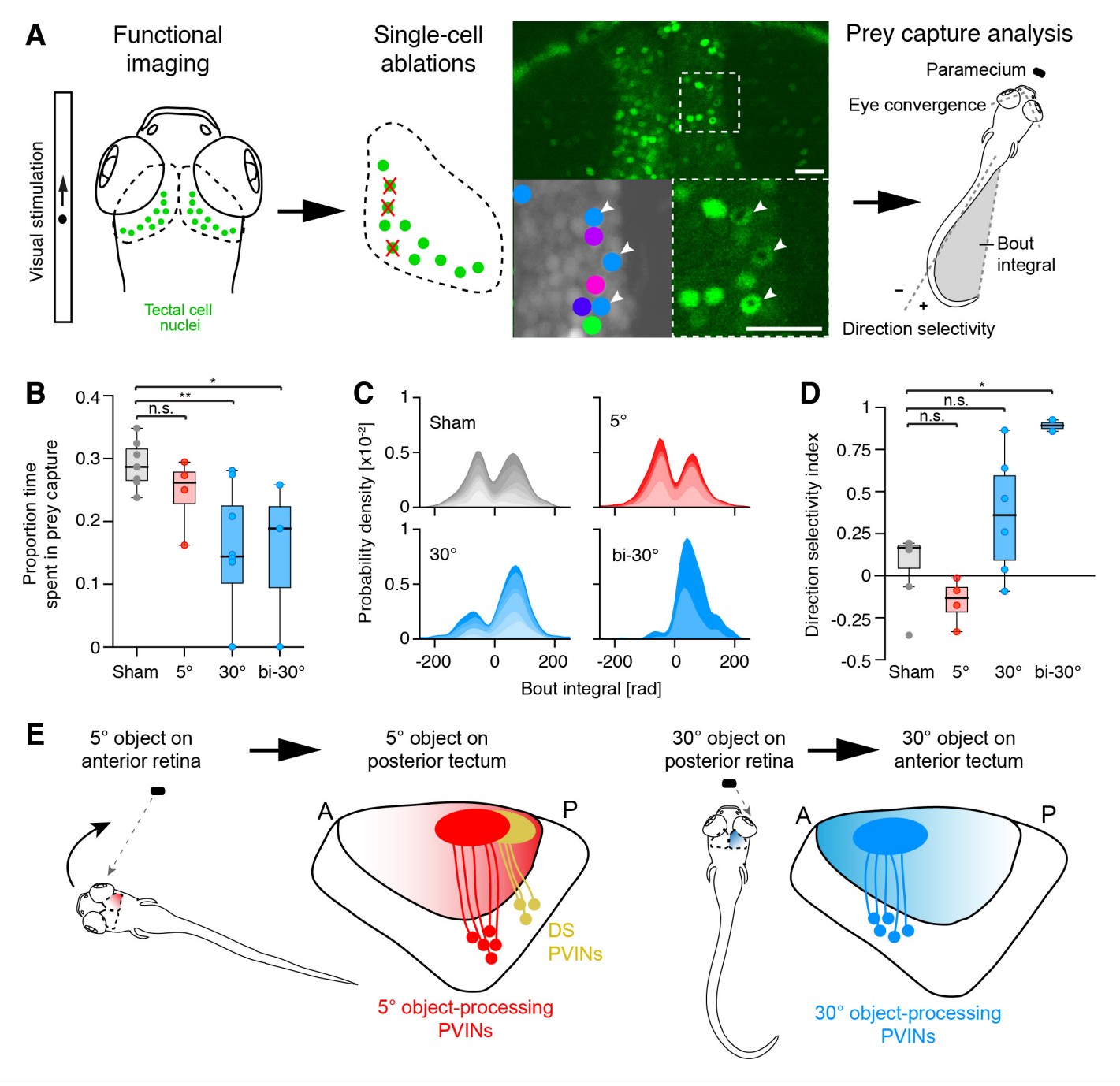

**Figure 8.** Large-object processing cells are required for hunting behavior. (**A**) 7 dpf old fish panneuronally expressing nuclear-localized GCaMP6s (green) were visually stimulated and imaged. Tectal cells were functionally identified (cluster-colored circles) and selected for ablations (arrowheads). At dpf, hunting behavior was analyzed in free-swimming larvae. (**B**) Proportion of time larvae spent engaged in hunting behavior, having their eyes converged. Single data points represent individual fish. 'Sham' (gray): control larvae with ablations of non-responsive cells, '5°' (red): unilateral ablations of 5°-dot-responsive cells in the right tectum, '30°' (blue): unilateral ablations of 30°-dot-responsive cells in the right tectum, 'bi-30°': bilateral ablations of 30°-dot-responsive cells. *: p = 0.02, **: p = 0.006, n.s.: p = 0.15. (**C**) Probability density plots of bout integrals for the initial J-turns, with positive values indicating a rightward and negative values indicating a leftward turn. Color shading indicates accumulated data for individual fish. (**D**) Direction selectivity index for initial J-turns of individual fish. *: p = 0.029, n.s.: p > 0.05. (**E**) Ethological relevance for A-P distribution of functionally distinct tectal cells. Before initiation of prey capture behavior, small moving objects are likely spotted in the temporal, monocular visual field. Precise recognition and processing of object movement by posterior DS cells avoids losing the object and enables adapted orientation turns towards the object. During prey pursuit, prey size seemingly increases and is detected by large-dot-responsive cells in the anterior tectum. Eye convergence allows binocular processing of object size and movement.

*Figure 8 continued on next page*

*Figure 8 continued*

The online version of this article includes the following figure supplement(s) for figure 8:

**Figure supplement 1.** Tectal cell ablation controls and enucleation experiments.

significant behavioral changes (*Figure 8B–D*). This suggests that for cells, which tile the visual field by only 5°, the ablated cell numbers were not sufficient to observe an effect on behavior.

## Both tectal hemispheres cooperate in guiding capture of prey in the frontal visual field

Intriguingly, we observed in our imaging experiments, that a substantial number of cells in the left tectum were responsive to prey-like stimuli presented to the left (ipsilateral) eye (*Figure 8—figure supplement 1D*). These cells are probably activated by the right (contralateral) tectum via an inter-tectal commissure. We hypothesized that these cells help to sharpen responses across both tecta by suppressing background activity in the tectum that is not directly stimulated by RGC inputs. To test this hypothesis, we laser-ablated large-dot responsive cells in the anterior tectum on both sides (*Figure 8B–D*). (Note that these cells were identified by imaging responses in both tecta to stimulation of only the left eye.) In bilaterally ablated animals, the tendency to increase right J-turns and reduce left J-turns in response to prey was even more pronounced than in right-tectum-only ablated larvae, supporting our hypothesis.

Background suppression of the tectal activity ipsilateral to the stimulated eye might enhance activity in the contralateral, visually stimulated tectum. To investigate this possibility, we imaged the tectum of fish in which the right eye was removed (*Figure 8—figure supplement 1E*). In these animals, we observed a significant increase in the number of large-dot responsive cells in the right, visually stimulated tectum, that is ipsilateral to the enucleated side. This result suggests that stimulus-evoked activity is normally dampened by background activity in the contralateral tectum by intertectal inhibitory connections. This background activity is suppressed, either physiologically by strong unilateral, stimulus-evoked activation of the other tectum (*Figure 8C and D*), or experimentally by removal of its own retinal inputs (*Figure 8—figure supplement 1E*). Taken together, our ablation results begin to reveal the logic of intertectal coordination of responses to prey in the frontal visual field.

## Discussion

In this study, we have discovered how the topographic layout of retinotectal circuitry is adapted to demands of the zebrafish larva's behavioral ecology. We postulate that natural selection has favored the evolution of position-dependent specializations in the neural architecture underlying the processing of object motion as it is caused by both the prey's and the fish's movements. The tectum is critically involved in identification, localization, pursuit, and capture of prey (*Gahtan et al., 2005*; *Semmelhack et al., 2014*). In the following, we will go through circuit adaptations to each of these functions.

Identifying prey, and distinguishing it from a potential predator, is critical for the larva's survival. In previous studies, we discovered that this distinction is made by size and movement characteristics of the perceived object (*Barker and Baier, 2015*). Small, sideways-moving dots are readily approached (*Semmelhack et al., 2014*), whereas expanding (looming) dots, displayed to the side of the fish while it is immobilized, are categorized as threatening and avoided by vigorous escape attempts (*Bhattacharyya et al., 2017*; *Temizer et al., 2015*). Here, we show that the RGC axon populations that respond to these two categories terminate in different layers of the tectum. Retinal inputs carrying prey-like signals mainly enter the tectum in layers SO and SFGS1-4, and looming-sensitive RGC axons are largely restricted to SFGS5/6 and SGC. In addition, broadly-tuned OFF signals are transmitted to the deep retinorecipient layers SGC and SAC/SPV. These include sudden and gradual transitions from light to dark.

The tectal cells that respond to these stimulus categories exhibit morphologies that match their predicted input channels, as previously shown for direction-selective RGC inputs and tectal cell dendritic arborizations (*Gabriel et al., 2012*). Prey-selective neurons extend dendrite branches into the

superficial layers of the tectum, while looming-sensitive neurons tend to arborize in middle to deep layers. As a general principle, most of the feature selectivities of tectal neurons are inherited from their functionally diverse RGC inputs. A simple excitation-only, feed-forward model showed that more than a third of the tectal response classes match a single RGC input class. The vast majority of the remaining responses were explained by a linear combination of two, or sometimes more, RGC inputs. In one experimentally accessible case, we could directly show that RGCs pass their small-dot responsive tuning on to downstream tectal cells. A similar modeling approach was recently performed to study functional connectivity between RGCs and the dorsolateral geniculate nucleus (dLGN) in the mouse thalamus (*Román Rosón et al., 2019*). Analogous to our findings, the authors described a high correlation between functional dLGN in- and output, and thus a low level of signal convergence.

The zebrafish tectum, however, is not merely a passive relay station for retinal inputs. First, we found that responses to large objects are markedly reduced in the tectum compared to RGCs. Second, direction selectivity to backward-moving objects is calculated de novo in the tectum. This was especially striking for the 5° dot stimulus. A circuit involving feed-forward inhibition by SINs, which suppresses tectal responses to non-preferred directions, could account for this computation (*Abbas et al., 2017*). Third, a substantial number of tectal cells selectively respond to a dark ramp stimulus; such cells were not observed in our RGC dataset. Thus, tectum-intrinsic circuitry adds direction selectivity to a subset of channels and generally refines and sharpens the responses.

The anatomical separation of small-dot responsive and looming-sensitive circuits probably reflects functional segregation of the two processing streams. *Barker and Baier, 2015* postulated a circuit motif that implements balanced, reciprocal inhibition of the two systems driving approach vs. avoidance. Such a circuit could generate a winner-take-all mechanism capable of coordinating behavioral responses to stimuli of opposite valence. The visual system needs to rapidly distinguish between prey and threat across the entire visual field. A specialization of tectal layers for the processing of key features orthogonal to the two retinotopic axes, as reported here, seems to be an adaptive solution for that challenge. Moreover, bundling in space the visual processing of object valence, global patterns and luminance levels by laminar separation may also serve to minimize wiring lengths of the corresponding neural elements in the tectal neuropil (*Baier, 2013*; *Chklovskii et al., 2002*).

By sampling feature-selective responses along the anterior-posterior axis of the tectum, we uncovered functional specializations of tectal regions, which probably reflect systematic changes in cell-type composition and connectivity. Object translation across the visual field is caused by a combination of both the prey's movement and the fish's own swimming, the latter often in response to position of the prey. Larval zebrafish are able to detect a prey item at a distance of several millimeters. A typical prey object, such as a paramecium or a rotifer, of 250 µm length, which is 3 mm away, subtends a visual angle of approximately 5°. Previously, we and others had detected responses of head-fixed larvae, embedded in agarose, to virtual, high-contrast objects of 2–6° diameter (*Bianco et al., 2011*; *Semmelhack et al., 2014*). Moving dots of 1° rarely elicited a response. This seems to be the resolution limit of the larval fish's visual system and is in agreement with the physical limit posed by photoreceptor spacing in the retina (*Haug et al., 2010*). As the fish turns toward and approaches the prey, the prey 'image' slides from nasal to temporal zones of the retina and from posterior to anterior regions of the tectum (*Figure 8E*). At the same time, the visual angle covered by the prey gradually increases. This might explain a shared sensitivity to slow-looming stimuli, which is featured by more than half of all large dot-responsive cells. Interestingly, this overlap is negligible for fast looming stimuli, what might indicate a separation of approach and avoidance circuits. The fish executes a capture strike when the prey is in the upper central field of both eyes at a distance range of 0.3–0.7 mm (*Mearns et al., 2020*). This corresponds to 20–40° of visual angle. Two tasks of successful hunting, the detection of distant prey in the peripheral visual field and the fixation of prey at close range in front of the animal shortly before the capture strike, informed our choice of 5° ('small') and 30° ('large') virtual objects for our imaging experiments. We discovered two asymmetries in the retinotectal map that appear to support these two different phases of hunting behavior.

First, an overrepresentation of small-dot responsive, direction-selective cells in the posterior tectum appears to be an adaptation to the preponderance of small prey objects in the peripheral field of view, whose movement is, at least initially, independent of the fish's own. These tectal cells acquire their direction selectivity by de novo computations from size-tuned, non-DS RGC inputs (*Figure 8E*). Determining the direction of prey by a local mechanism is particularly important for the

lateral field of view, which is entirely monocular. The further away from the midline the prey's location is the greater the turning angle that is needed to steer the fish toward its food. Fish will preferentially orient towards prey in their lateral visual field, because it gives them more time to move their body into the right position for a successful strike. Moreover, turning is energetically costly and may alert nearby predators and prey alike. For a prey object that already moves from back to front and whose image therefore slides from the nasal to the temporal retina, the turn angle will be smaller: the food may swim right in front of the fish, from where it might even be sucked into the mouth without extensive pursuit. Zebrafish larvae have been observed to use such a sit-and-wait mode of hunting (*Patterson et al., 2013*).

Second, the anterior tectum is enriched for large-dot responsive tectal cells (*Figure 8E*), which appear to facilitate the initiation of prey capture-associated J-turns, as shown here by laser ablation. J-turns are fine adjustments of body posture characteristic of hunting. These cells are frequently not direction-selective and communicate via commissural connections with the contralateral tectum. Initial imaging and behavioral experiments following ablations suggest that activation of large-dot selective cells suppresses responses in the contralateral tectum. We propose that such an intertectal inhibitory mechanism helps to correct slight displacements of the prey from the midline (see also *Gebhardt et al., 2019*). This signal may be transformed into fine orienting tail movements by one of the tectorecipient premotor areas in the hindbrain (*Helmbrecht et al., 2018*).

In conclusion, this work has revealed a neural architecture of the tectum that is well adapted to the demands of the animal's behavioral ecology. More generally, we demonstrate that the well-studied retinotectal map is spatially organized by function along both its retinotopic and laminar axes. The visual map in the tectum is thus not a veridical, unbiased representation of all positions in visible space, but rather warped by location-dependent feature statistics. Future work will undoubtedly uncover additional adaptations and will shed light on both the proximate, developmental mechanisms and the ultimate, evolutionary forces that are shaping this important visuomotor hub in the vertebrate brain.

# Materials and methods

## Key resources table

| Reagent type (species) or resource | Designation | Source or reference | Identifiers | Additional information |
|---|---|---|---|---|
| Chemical compound, drug | Alpha-Bungarotoxin | Invitrogen | Invitrogen:B1601 | |
| Chemical compound, drug | Tricaine | Sigma-Aldrich | Sigma-Aldrich :MS-222 | |
| Genetic reagent (*Danio rerio*) | Tg(elavl3:nls-GCaMP6s)mpn400 | *Förster et al., 2017* | ZFIN ID: ZDB-ALT-170731-37 | |
| Genetic reagent (*Danio rerio*) | Tg(atoh7:Gal4-VP16)s 1992t (ath5:Gal4) | *Del Bene et al., 2010* | ZFIN ID: ZDB-FISH-150901-27082 | |
| Genetic reagent (*Danio rerio*) | Tg(UAS:GCaMP6s)mpn101 | *Thiele et al., 2014* | ZFIN ID: ZDB-FISH-150901-22562 | |
| Genetic reagent (*Danio rerio*) | Et(E1b:Gal4-VP16)s1101t | *Scott et al., 2007* | ZFIN ID: ZDB-FISH-150901-5255 | |
| Genetic reagent (*Danio rerio*) | Tg(elavl3:lyn-tagRFP)mpn404 | *Dal Maschio et al., 2017* | ZFIN ID: ZDB-ALT-170731-38 | |
| Genetic reagent (*Danio rerio*) | Tg(isl2b:Gal4-VP16, myl7:TagRFP)zc65 | *Fujimoto et al., 2011* | ZFIN ID: ZDB-FISH-150901-13523 | |
| Genetic reagent (*Danio rerio*) | Tg(14xUAS:EGFP)mpn100 | *Thiele et al., 2014* | ZFIN ID: ZDB-GENO-140812-1 | |
| Genetic reagent (*Danio rerio*) | Tg(Shha:GFP)t10 | *Neumann and Nuesslein -Volhard, 2000* | ZFIN ID: ZDB-GENO-060207-1 | |
| Genetic reagent (*Danio rerio*) | Tg(UAS:mCherry)s1984t | *Heap et al., 2013* | ZFIN ID: ZDB-FISH-150901-14417 | |

*Continued on next page*

*Continued*

| Reagent type (species) or resource | Designation | Source or reference | Identifiers | Additional information |
|---|---|---|---|---|
| Genetic reagent (*Danio rerio*) | Tg(brn3c:GAL4, UAS:gap43-GFP)s318t (BGUG) | *Xiao and Baier, 2007* | ZFIN ID: ZDB-ALT-070423-6 | |
| Genetic reagent (*Danio rerio*) | Tg(UAS:paGFP,nlsG CaMP6f)mpn104 (UAS:FuGIMA-f) | This paper | | Tol2-mediated transgenesis |
| Software, algorithm | Imaris | Bitplane | | http://www.bitplane.com |
| Software, algorithm | ImageJ/Fiji | *Schindelin et al., 2012* | | http://fiji.sc |
| Software, algorithm | MorphoLibJ (ImageJ plugin) | *Legland et al., 2016* | | http://imagej.net/morpholibj |
| Software, algorithm | PsychoPy2 | *Peirce, 2007* | | http://www.psychopy.org |
| Software, algorithm | Python 2.7 | Python.org | | http://www.python.org |
| Software, algorithm | Python 3 | Python.org | | http://www.python.org |
| Software, algorithm | CalmAn (Calcium Imaging Analysis toolbox) | *Giovannucci et al., 2017* | | http://github.com/flatironinstitute/CaImAn |
| Software, algorithm | NeuTube | *Feng et al., 2015* | | http://www.neutracing.com |
| Software, algorithm | Advanced Normalization Tools (ANTs) | *Avants et al., 2010* | | http://stnava.github.io/ANTs |
| Software, algorithm | RStudio Version 1.0.136 | RStudio | | http://www.rstudio.com |
| Software, algorithm | R package nat (NeuroAnatomy Toolbox) | *Bates et al., 2020* | | http://jefferis.github.io/nat/ |
| Software, algorithm | 3DSlicer | *Fedorov et al., 2012* | | http://www.slicer.org |
| Software, algorithm | Plotly Chart Studio | Plotly.com | | http://www.plotly.com |
| Software, algorithm | Custom tracking and behavior analysis code | *Mearns et al., 2020* | | http://bitbucket.org/mpinbaierlab/mearns_et_al_2019 |

## Experimental model and subject details

All animal procedures conformed to the institutional guidelines set by the Max Planck Society, and were approved by the regional government of Upper Bavaria (Regierung von Oberbayern; approved protocols: ROB-55.2-1-54-2532-101-2012 and ROB-55.2–2532.Vet_02-19-16).

## Transgenic constructs

To generate *UAS:FuGIMA-f*, paGFP (gift from K. Svoboda, addgene no. 18697) and nls-GCaMP6f (*Förster et al., 2017*) were cloned on either side of a bidirectional 14xUAS in a Tol2 vector, featuring a transgenesis marker ('bleeding heart', *cmlc2:mCherry*). Transgenic fish were generated using the standard Tol2 transposon system, and the highly variegated line *Tg(UAS:paGFP,nlsGCaMP6f) mpn104* was used for experiments.

## Transgenic zebrafish lines

For all experiments, we used 5–7 days post fertilization (dpf) larvae carrying mutations in the *mitfa* gene (*nacre*), which were raised on a 14 hr light/10 hr dark cycle at 28°C. To record functional responses to visual stimuli of tectal cells, we used *Tg(elavl3:nls-GCaMP6s)mpn400* fish and similarly for RGCs, we used *Tg(atoh7:Gal4-VP16)s1992t; Tg(UAS:GCaMP6s)mpn101* fish. RGC axon ablation experiments were performed in *Tg(atoh7:Gal4-VP16)s1992t; Tg(UAS:mCherry)s1984t; Tg(elavl3:nls-GCaMP6s)mpn400* fish, and tectal cells were ablated in *Tg(elavl3:nls-GCaMP6s)mpn400* fish.

FuGIMA experiments were performed in incrossed *Et(E1b:Gal4-VP16)s1101t (=Gal4^s1101t); Tg (UAS:paGFP,nlsGCaMP6f)mpn104 (=FuGIMA-f); Tg(elavl3:lyn-tagRFP)mpn404* fish. Other single-cell reconstructions were generated using *Et(E1b:Gal4-VP16)s1013t (=Gal4^s1013t); Tg(brn3c:Gal4, UAS: gap43-GFP)s318t (=BGUG)* fish. To define tectal layers, RGC expression in *Tg(isl2b:Gal4-VP16)zc65; Tg(14xUAS:EGFP)mpn100* fish and in *Tg(Shha:GFP)t10* fish was used. To allow registrations to a standard brain, all fish were crossed to the line *Tg(elavl3:lyn-tagRFP)mpn404*.

## Tectal cell counts

5–7 dpf larvae expressing *elavl3:nls-GCaMP6s* were embedded in 2% low-melting-point agarose and a lethal dose of tricaine methanesulfonate (MS-222) was applied. After 15 min, the tectal brain regions were imaged on a Zeiss LSM780 microscope (voxel size: $0.27 \times 0.27 \times 1.5$ µm$^3$). Images were manually segmented in Imaris (v8.0, Bitplane) by setting pixel intensities outside of the tectum to 0. Using ImageJ (v1.52n), pixel intensities were inverted, images were Gaussian filtered and a classic watershed segmentation was applied (MorphoLibJ plugin). ROIs smaller than 400 voxels were removed and the number of ROIs was analyzed in 3D.

## Functional imaging and visual stimulation

In vivo calcium imaging was performed on a previously described two-photon microscope (*Förster et al., 2017*) on 5–7 dpf transgenic zebrafish larvae expressing either cytoplasmic GCaMP6s in RGCs or nuclear-localized GCaMP6s panneuronally. Larvae were mounted in 2% low-melting-point agarose. The stimulus was projected onto a white diffusive screen using the red channel of a LED projector, in a distance of 4 cm from the larva. The projection was presented monocularly and covered ~120° of the larva's field of view. GCaMP6 signals were recorded by scanning at 920 nm, at ~2 Hz, at a resolution of ~0.6 µm/pixel. The tectum was covered in depth by acquiring z-planes with a distance of ~7 µm.

Visual stimulation was designed using PsychoPy2 and consisted of a dark ramp (red to black, 3 s), a bright ramp (black to red, 3 s), a dark flash (red to black), and a bright flash (black to red). This was followed by a small horizontally moving dot (5°, 90°/s) in forward (temporal to nasal) and backward (nasal to temporal) directions (two repetitions each), and at two elevations of the screen, first at equatorial plane and then elevated by ~20° (two repetitions each). We chose dark dots on a bright (red) background. Published (*Antinucci et al., 2019*) and our own unpublished results have shown that these stimuli are efficient at eliciting hunting-like behavior in a dark 2P microscope environment, in the absence of UV stimulation (*Yoshimatsu et al., 2020*). Subsequently, a big dot was moving horizontally (30°, 90°/s) in forward and backward directions (repeated twice), at an elevation of ~10°, thus covering the two horizontal planes of the small dot. The frequency control consisted of black gratings with a spatial frequency of 5° and a temporal frequency of 90°/s, moving in forward and backward directions (repeated twice). The looming stimuli consisted of a fast (~60°/s, linear expansion) and a slow-looming disc (~20°/s, linear expansion), both ending with a black screen (two repetitions each). This stimulus protocol was repeated twice with a total acquisition length of 515 s.

## Analysis of imaging data

Recorded imaging data were pre-processed as described previously (*Helmbrecht et al., 2018*). In brief, images were motion-corrected using the CaImAn package, uniformly filtered over three frames and the dF/F was calculated using the 5$^{th}$ percentile of the traces. In total 15 regressors for all stimulus components were created and convolved with a corresponding GCaMP6 kernel. Neuronal activity was analyzed pixel-wise for RGC and ROI-wise for tectal imaging data, by calculating a score of all regressors to the calcium responses of each pixel using a linear regression model of the selected response window with the regressor (Python scikit-learn). For the score, the coefficient of the regression (CR; corresponding to the dF/F) was multiplied by the correlation value $R^2$. All pixels and ROIs were imaged twice using the same stimulus and the final score was calculated via a weighted average of the scores by the corresponding $R^2$.

## Clustering of functional responses

To determine overall response types, the scores were normalized per fish to the 99$^{th}$ percentile of all pixels/ROIs recorded.

For the functional clustering of the responsive tectal ROIs, three fish (7594 ROIs) expressing *elavl3:nls-GCaMP6s* were analyzed by first removing ROIs with maximum scores smaller than 0.2 (1908 ROIs remaining). Next, to reduce noise and to find local structure in the dataset, affinity propagation clustering (scikit learn – preference: median of similarities) was performed (151 clusters). Keeping clusters with at least 5 ROIs, yielded in total 80 clusters with chosen exemplars. To extract the global cluster structure, these 80 exemplars were further clustered using hierarchical clustering (scipy.cluster) using correlation as distance metric. Clusters with less than 20 ROIs were removed. We calculated a silhouette coefficient to validate the clustering. A distance threshold of 0.25 was chosen, which yielded a minimal number of clusters (14) with the highest silhouette coefficient. This finally resulted in 14 tectal cell clusters with a total of 76 exemplars and 1759 ROIs (92.2%). Principal component analysis (PCA) was performed on the score values of each stimulus for all tectal cells.

Similarly to the clustering of tectal neurons, the responsive RGC pixels of one fish (14 planes; each 297 × 303 pixel) expressing *ath5:Gal4 UAS:GCaMP6s* were analyzed by again removing pixels with maximum scores smaller than 0.4 (remaining 58,910 pixel) and performing affinity propagation clustering (scikit learn – preference: median of similarities). Keeping clusters with at least five pixels (0.01% of all pixels), yielded in total 1243 clusters with chosen exemplars. These 1243 exemplars were further ordered by hierarchical clustering (scipy.cluster) using correlation as distance metric. Cluster with less than 589 pixels (1% of all pixels) were removed. After silhouette analysis, a distance threshold of 0.45 was chosen, which yielded ten clusters with a total of 1157 exemplars and 55,153 pixels (93.6%). Although four RGC clusters yielded a higher silhouette coefficient, we chose ten clusters, which resulted in a significantly higher correlation value ($R^2_{score}$) for the following linear modeling analysis (see *Figure 4—figure supplement 1A*).

To quantify the number of pixels per RGC cluster in tectal compartments and layers (*Figure 3C* and *Figure 3—figure supplement 1D*), we used ImageJ to manually draw ROIs and to count pixels for each compartment/lamina in each image plane.

## Mapping of functional responses from independent experiments onto our clustered datasets

To map response types of SINs, NINs and enucleated fish, functional imaging was performed as described. ROIs were defined semi-automatically to segment only single, separated tectal cell bodies in the tectal neuropil and/or the periventricular layer. Several fish per experiment were analyzed to calculate the scores, and again pixels with maximum scores smaller than 0.2 were removed. A k-nearest neighbor classifier (`sklearn.neighbors.KNeighborsClassifier`) was trained on the *elavl3:nls-GCaMP6s* clustered ROIs (1759 ROIs with cluster labels, k = 10) and the scores of every mapped fish were assigned to the cluster dataset using either predicted labels for the ROIs distribution or probability estimates for the population distributions. The classification was cross-validated by splitting the *elavl3:nls-GCaMP6s* dataset into 70% training and 30% test data, which evaluated to an accuracy of 92%. A similar, pixel-wise approach was used to map the functional RGC data of two additional *ath5:Gal4; UAS:GCaMP6s* fish onto the ten RGC clusters by choosing k = 100 (*Figure 3—figure supplement 1C*).

## Modeling of tectal responses using RGC inputs

To predict the tectal responses using RGC information, we applied a linear modeling approach using L1-regularized regression (Lasso) (`sklearn.linear_model.Lasso`) with non-negative constraint. The cost function of the Lasso is defined by:

$$Cost = \sum_{i=0}^{n} \left( yi - \sum_{j=0}^{m} wj\, xij \right)^2 + \lambda \sum_{j=0}^{m} |wj|$$

The regularization parameter ($\lambda$) helps to reduce the impact of multicollinearities between the average scores of RGC classes, and the optimal $\lambda$ was found by minimizing the mean squared error of a grid search on a log scale between $1e^{-5}$ and $1e^{-1}$ (*Figure 4—figure supplement 1B*). The modeling of the scores of every single tectal neuron (total 1759) was performed using the 15-dimensional average scores of the 10 defined RGC clusters, so that:

$$PredScore\ TectalNeuron = b + \sum_{j=0}^{m\,(RGC)} wj\ AvgScoreRGCj$$

The *PredScore* was evaluated by calculating the $R^2_{score}$ of the regression. To predict the calcium traces of the tectal cells, we used the resulting weights of the regression and calculated the dot product of the average RGC responses with the corresponding weights (w) and bias (b) and evaluated the result via the pearson correlation ($R_{trace}$) between the predicted and measured calcium responses.

The model was tested by comparing the resulted distribution of response correlations to the distribution of a random model, by choosing for every cell 1000 times random weights (*Figure 4—figure supplement 1C*). In addition, the model was cross-validated by splitting the data into a training and test set using one of the two trials per cell, and a corresponding RMSE (root mean squared error) of the test dataset was calculated (*Figure 4B*).

## FuGIMA and other single-cell labeling experiments

Tectal responses in fish expressing *elavl3:lyn-tagRFP* and *UAS:FuGIMA-f* under control of *Gal4s1101t* were functionally imaged as described above. After image acquisition, a custom-written, regressor-based python script was used to overlay a color map of correlated pixels on the mean ΔF/F image to identify cells of functional interest. Single-cell photoactivation of paGFP was performed as previously described (*Förster et al., 2018*). Typically, 2–3 photoactivation cycles were sufficient to reach the maximal fluorescence intensity in tectal interneurons. After allowing paGFP to diffuse into all neurites of the photoactivated cell for about 30–45 min, a high-resolution z-stack of the whole tectum, including both paGFP and lyn-tagRFP channels, was acquired at a confocal microscope (LSM700 or LSM780, Zeiss; 20x/1.0 NA water-dipping objective).

Other single-cell reconstructions (randomly-labeled tectal neurons) were performed using the BGUG method as previously published (*Helmbrecht et al., 2018*). In brief, fish expressing a highly variegated Gap43-GFP under control of the tectal *Gal4s1013t* line were crossed to *elavl3:lyn-tagRFP* fish and offspring were screened for sparse GFP expression in tectal interneurons.

All individual neurons were traced semi-automatically using the software neuTube (Build1.0z) and SWC files were generated for each cell.

## Image registration

All image registrations were performed using the Advanced Normalization Tools (ANTs) software (*Avants et al., 2010*), and live expression of *elavl3:lyn-tagRFP* served as a reference channel. First, a FuGIMA standard brain was generated by mirroring all FuGIMA cells to one brain half and by subsequent registration to one exemplary lyn-tagRFP channel, which served as a template. ANTs parameters recently determined for live samples were applied (*Marquart et al., 2017*). Second, this FuGIMA standard brain was registered to the zebrafish single-neuron atlas (*Kunst et al., 2019*) in three steps: (1) registration of the FuGIMA template to a tectal subvolume of the live lyn-tagRFP standard brain from the atlas, (2) extension to the full live standard brain volume, (3) registration of the live standard brain to the fixed standard brain of the atlas. Similarly, the BGUG dataset was first registered to its own standard brain, which was subsequently registered to the single-neuron atlas. Finally, single-neuron tracings (SWC files) were aligned using the antsApplyTransformToPoints function contained in the ANTsR package. For visualizations and 3D renderings, we used the web interface of the single-neuron atlas (http://fishatlas.neuro.mpg.de/). All single-neuron data from this study are publicly available through this atlas.

## Morphological quantifications

To add landmarks for the tectal laminae, we co-registered the expression patterns of *isl2b:Gal4 UAS:GFP* and *shh:GFP* into the FuGIMA standard brain. We then used these anatomical labels, together with the software 3D slicer (http://www.slicer.org/), to manually segment the individual tectal layers. For every cell, we measured the fiber lengths in each layer and calculated the percentage of the cell's total neurite length (proportional branch length) using a custom-written python script. Single-cell morphological barcodes (heatmaps) were generated using Plotly Chart Studio (https://plot.ly/).

To quantify the neurite arbor size of tectal cells, we used the 'Oblique slicer' and 'Measurement points' tools in Imaris (v8.02; Bitplane) to define and extract planar coordinates for each laminar stratification (*Figure 5—figure supplement 3A*). The areas in µm$^2$ were quantified using a custom-written python script.

## Ablation and enucleation experiments

For RGC axon ablations, 6 dpf old larvae expressing mCherry in RGCs and nuclear GCaMP6s pan-neuronally were mounted in agarose and were intraspinally injected with alpha-bungarotoxin (2 mg/ml, Invitrogen, B1601). Tectal cell responses were functionally imaged as described above. Subsequently, the axon bundle, which leaves AF7 for the tectal SO layer was cut at the same 2P microscope by scanning a 10 µm line (0.01 µm/pixel) at 760 nm for 500 ms transverse to the fascicle. The laser intensity at the objective focal plane was ~30 mW. Afterwards, fish were released from agarose to recover overnight in Danieau's solution. At 7 dpf, fish were re-embedded and functional imaging of tectal cell responses was repeated. Somata signals in the tectal neuropil served as landmarks for approximate reidentification of the same imaging planes obtained at 6 dpf. Regressor analysis was described as above and cluster-color-coded responsive tectal cells were counted manually.

For tectal cell ablations, 7 dpf old larvae expressing nuclear GCaMP6s panneuronally were embedded in agarose and functionally imaged at the 2P microscope. Up to 3–5 tectal cells per imaging plane (max. 15 cells per fish) were selected for their response type, and were ablated by 30 ms two-photon laser pulses (800 nm, ~35 mW), pointed at the nucleus.

For enucleation experiments, 4 dpf old fish expressing *elavl3:nls-GCaMP6s* were placed in 2% low-melting agarose with 0.02% tricaine methanesulfonate (MS-222). The right eye was removed using custom-made micro-scalpels. Fish were allowed to recover for two days in Danieau's solution until functional imaging was performed at 6 dpf.

## Free-swimming prey capture assay

Prior to testing prey capture behavior, larvae were allowed to feed ad libitum on paramecia from 5 to 6dpf. At 7 dpf, larvae were embedded in agarose and cells in the tectum were ablated (see above). Larvae were freed from agarose and allowed to recover overnight. Prey capture behavior was tested the following day at 8 dpf. Controls groups were unembedded siblings, siblings embedded but not subject to the ablation protocol, and 'sham' ablated siblings.

The free-swimming prey capture assay was performed as described previously (*Mearns et al., 2020*). Briefly, larvae were introduced individually into an arena (15 × 15 × 5 mm) with 50–100 paramecia (*Paramecium multimicronucleatum*). Each larva was allowed to feed for 20–30 min while being recorded from above at 500 frames per second using a high-speed camera (PhotonFocus, MV1-D1312-160-CL, Switzerland). In each frame of the recordings, the eyes and tail of the fish were tracked offline using custom-written Python software (https://bitbucket.org/mpinbaierlab/mearns_et_al_2019). Tail tracking was performed using background subtraction and thresholding followed by skeletonization of the largest contour in the image. Swim bouts were identified using a change point algorithm on the derivative of the tail angle with respect to time. Eye tracking was performed similarly using background subtraction, thresholding and contour detection. For each animal independently, we calculated the distribution of eye convergence angles over the experiment and used the local minimum in the resulting bimodal distribution as the prey capture threshold. Since eye convergence is a reliable indicator of prey capture in zebrafish larvae (*Bianco et al., 2011*; *Patterson et al., 2013*; *Mearns et al., 2020*), we defined hunting events as any time the eye convergence angle was above this threshold. Initial orienting J-turns were defined as any bout where the eyes were unconverged before and converged after the bout. The bout integral was calculated by summing the tail tip angle values over the duration of the bout, with positive values indicating a rightward turn and negative values indicating a leftward turn. The direction of the turn was defined by the sign of the bout integral (positive for right, negative for left). The direction selectivity index was computed as [(# right J-turns - # left J-turns) / (total # J-turns)] for each fish, with a value of 1 indicating all J-turns were to the right, −1 indicating all J-turns were to the left, and 0 indicating no overall bias in J-turn direction.

## Statistical analysis

Statistical tests were two-tailed t-tests, if not stated otherwise. For the quantification of prey capture behavior, statistics were performed using the scipy library in Python 3. The proportion of time larvae spent engaged in hunting behavior was compared between treatment groups using a Mann-Whitney U test. Similarly, the direction selectivity index of initial J-turns was compared between treatment groups using a Mann-Whitney U test.

## Acknowledgements

We thank Irene Arnold-Ammer for help with molecular biology, Krasimir Slanchev for support during fish maintenance and husbandry, Michael Kunst for advice on image registration and the entire Baier lab for constructive feedback and discussions throughout the project. This study made use of the high-performance application services at the Max Planck Computing and Data Facility for processing image registrations. Funding was provided by the Max Planck Society (all authors).

## Additional information

### Funding

| Funder | Author |
| --- | --- |
| Max Planck Society | Dominique Förster<br>Thomas O Helmbrecht<br>Duncan S Mearns<br>Linda Jordan<br>Nouwar Mokayes<br>Herwig Baier |

The funders had no role in study design, data collection and interpretation, or the decision to submit the work for publication.

### Author contributions

Dominique Förster, Conceptualization, Data curation, Formal analysis, Methodology, Writing - original draft, Project administration, Designed the experiments. Performed the experiments with support from LJ; Thomas O Helmbrecht, Conceptualization, Data curation, Software, Formal analysis, Writing - review and editing, Designed the experiments, Analyzed the data; Duncan S Mearns, Data curation, Software, Methodology, Writing - review and editing, Performed and analyzed prey capture experiments; Linda Jordan, Data curation, Methodology; Nouwar Mokayes, Data curation, Software, Wrote the code for morphological quantifications; Herwig Baier, Conceptualization, Supervision, Funding acquisition, Writing - review and editing

### Author ORCIDs

Dominique Förster ⓘ https://orcid.org/0000-0002-7821-6755
Herwig Baier ⓘ https://orcid.org/0000-0002-7268-0469

### Ethics

Animal experimentation: All animal procedures conformed to the institutional guidelines set by the Max Planck Society, and were approved by the regional government of Upper Bavaria (Regierung von Oberbayern; approved protocols: ROB-55.2-1-54-2532-101-2012 and ROB-55.2-2532.Vet_02-19-16).

### Decision letter and Author response

Decision letter https://doi.org/10.7554/eLife.58596.sa1
Author response https://doi.org/10.7554/eLife.58596.sa2

## Additional files

### Supplementary files

- Transparent reporting form

### Data availability

All data generated or analysed during this study are included in the manuscript and supporting files. Single cell reconstructions are available through our zebrafish atlas at http://fishatlas.neuro.mpg.de/.

The following datasets were generated:

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
