## [Decision Letter]

**Acceptance summary:**

The authors analyzed responses of tectal cells to prey-like visual stimuli by combining 2P imaging, photoactivatable GFP, and ablation and behavioral experiments, and elegantly demonstrate that the feature selectivity and connections of neurons in the optic tectum may be adapted to the detection of prey swimming across the visual field. Specifically, they show that the topographic organization of retinotectal circuitry is well matched to the behavioral ecology of the fish, establishing a possible link between specific functional/anatomical circuits and processing of the visual features of potential prey.

**Decision letter after peer review:**

Thank you for submitting your article "Retinotectal circuitry of larval zebrafish is adapted to detection and pursuit of prey" for consideration by *eLife*. Your article has been reviewed by three peer reviewers, including Koichi Kawakami as the Reviewing Editor and Reviewer #1, and the evaluation has been overseen by Andrew King as the Senior Editor. The following individual involved in review of your submission has agreed to reveal their identity: Jiulin Du (Reviewer #3).

The reviewers have discussed the reviews with one another and the Reviewing Editor has drafted this decision to help you prepare a revised submission.

Our expectation is that the authors will eventually carry out the additional experiments (if the authors think necessary) and report on how they affect the relevant conclusions either in a preprint on bioRxiv or medRxiv, or if appropriate, as a Research Advance in *eLife*, either of which would be linked to the original paper.

Summary:

The authors analyzed responses of tectal cells to prey-like visual stimuli by combining 2P imaging, photoactivatable GFP and ablation and behavioral experiments, and discovered that posterior neurons respond to small objects and compute their direction of movement dependently on monocular vision, and anterior neurons are tuned to larger objects (not direction-selective) dependently on binocular vision. Interestingly, the tectal cells with different functionalities have characteristic morphology and arborizations specific to retinal ganglion cell synaptic inputs. Thus, the authors demonstrated that links between specific functional/anatomical circuits and perception of a prey. Overall, the experiments are well-designed and the results presented are clear and interesting. The manuscript was clearly written and easily understandable.

Major comments:

1) Citations are generally sparse and selective. For example, the first page of the Introduction (the first and second paragraphs ), which makes claims about efficient coding, natural vision, and corresponding neural organization, is unreferenced.

Of the first six references that appear on the next page, five are by the same lab as the present study. A lack of important citations is notable across the paper.

Introduction: "the position of an object in the visual field matches a corresponding focus of activity in tectal space" Especially this was clearly shown by calcium imaging by Muto et al., 2013. Also, in Muto et al., the correlation between activation of the anterior tectum and initiation of prey capture was described.

Wang et al., 2020, showed asymmetries in the encoding of different types of motion in the larval zebrafish tectum (not cited). This contrasts the author's assertion: "systematic changes in cell type composition of connectivity along the anterior-posterior or dorsal-ventral axes of the tecum, resulting in gradients of other asymmetries of feature selectivity, have to our knowledge not been reported".

In addition, the asymmetry is perhaps not all that surprising since RGCs coming out of the eye are already asymmetric (Robles et al., 2014; Zhou et al., 2020 bioRxiv). Clearly, the authors have gone beyond what was previously shown in the above papers, but it would be good to see a broader acknowledgement of previous work.

Moreover, some assertions might be seen as grandiose. For example, in the Introduction: "We posit that the local statistics of the sensory environment, […], shape topographic specializations of sensory and sensorimotor circuitry". One might argue that, with classic textbooks entirely devoted to this point, the time to posit this notion has passed.

2) The reviewers concern if the black object used in this study represents paramecia (prey capture event). They may be pieces of debris or similar that are poorly reflective and thus appear darker than background. Insect cuticle, for example, tends to be "black" under most viewing conditions. The following points should be considered and discussed.

It is unclear how strongly the stimulus battery used will activate prey-capture circuits:

a) In nature, paramecia are brighter than the background, however the schematics shown in Figure 1E indicate that dark spots were used as the stimulus (Materials and methods are unclear about this). The same lab previously used bright stimuli to study the same behavior with great success (e.g. Semmelhack et al., 2014), why then use presumably less behaviorally relevant dark ones here?

b) Stimulus sizes used (5 deg for small, 30 deg for large) seem quite large, and based on the assumption that zebrafish prey (paramecia) are 250 microns in length, which is very large for a paramecium, and it also implies that they are viewed from the side.

c) Stimuli were red and black, but paramecia are more visible at shorter wavelengths (e.g. Yoshimatsu et al., 2019 bioRxiv – a study "Yoshimatsu et al., 2020" is cited in the Introduction in a different context, but it is missing from the reference list).

Rather, equally interesting other visual circuits are probably driven more effectively with the stimuli as used. Would therefore the framing be better attempted around a characterization of size and motion vision in the early visual brain in general, which would be interesting and novel in its own right?

Bianco, 2011 e.g. showed that black objects "work". Unfortunately the authors insist on linking their findings specifically to paramecia, which simply are not "black". And they are also much smaller that suggested. If they want to link this data to prey capture, they should acknowledge this.

It is quite clear that prey detection performance is much higher when the prey appears in front of the fish – the authors themselves have shown this (Mearns et al, 2020), after earlier observations by others (e.g. Bianco, 2011). Specifically, zebrafish show best detection performance when the prey lines up with the acute zone, about 20-30 deg laterally offset, just above the horizon. For this reason e.g. Semmelhack et al., 2014 (also a paper from the authors) put the screen in front of the fish, not to the side. It therefore seems quite clear that the primary prey detection circuit sits in the retina's acute zone. This also makes much more sense – the acute zone has slightly better resolution that the rest of the eye.

---

## [Author Response]

Major comments:1) Citations are generally sparse and selective. For example, the first page of the Introduction (the first and second paragraphs ), which makes claims about efficient coding, natural vision, and corresponding neural organization, is unreferenced.Of the first six references that appear on the next page, five are by the same lab as the present study. A lack of important citations is notable across the paper.

The reviewers' concern is well taken. We have now added a substantial amount of additional citations throughout the whole paper.

Introduction: "the position of an object in the visual field matches a corresponding focus of activity in tectal space" Especially this was clearly shown by calcium imaging by Muto et al., 2013. Also, in Muto et al., the correlation between activation of the anterior tectum and initiation of prey capture was described.

This reference has been added to the Introduction.

Wang et al., 2020, showed asymmetries in the encoding of different types of motion in the larval zebrafish tectum (not cited). This contrasts the author's assertion: "systematic changes in cell type composition of connectivity along the anterior-posterior or dorsal-ventral axes of the tecum, resulting in gradients of other asymmetries of feature selectivity, have to our knowledge not been reported".

Wang et al. report an abundance of small size-sensitive tectal cells in the upper nasal visual field. This reference has been added to the Introduction.

In addition, the asymmetry is perhaps not all that surprising since RGCs coming out of the eye are already asymmetric (Robles et al., 2014; Zhou et al., 2020 bioRxiv). Clearly, the authors have gone beyond what was previously shown in the above papers, but it would be good to see a broader acknowledgement of previous work.

We added references to acknowledge previously identified asymmetric RGC type distribution in the zebrafish retina.

Moreover, some assertions might be seen as grandiose. For example, in the Introduction: "We posit that the local statistics of the sensory environment, […], shape topographic specializations of sensory and sensorimotor circuitry". One might argue that, with classic textbooks entirely devoted to this point, the time to posit this notion has passed.

We appreciate the reviewers' concern and toned down our statement.

2) The reviewers concern if the black object used in this study represents paramecia (prey capture event). They may be pieces of debris or similar that are poorly reflective and thus appear darker than background. Insect cuticle, for example, tends to be "black" under most viewing conditions. The following points should be considered and discussed.It is unclear how strongly the stimulus battery used will activate prey-capture circuits:a) In nature, paramecia are brighter than the background, however the schematics shown in Figure 1E indicate that dark spots were used as the stimulus (Materials and methods are unclear about this). The same lab previously used bright stimuli to study the same behavior with great success (e.g. Semmelhack et al., 2014), why then use presumably less behaviorally relevant dark ones here?

The reviewers are correct that paramecia appear brighter than the background under naturalistic conditions. However, the natural prey of zebrafish larvae in the wild is not known, and it is plausible that larvae also prey on items that are darker than the background. Indeed, the RGCs believed to underpin prey capture have dendritic arbors in both ON and OFF layers of the IPL and respond to OFF, as well as ON, stimuli (Semmelhack et al., 2014; Zhou and Bear et al., 2020). Under 2P imaging conditions, dark spots are just as efficient as bright spots at eliciting hunting-like behavior (Bianco and Engert, 2015; Antinucci et al., 2019; unpublished observations from our lab). It should be noted that in a dark 2P microscope environment larvae may be less efficient at detecting a bright red dot since shorter wavelengths, which are believed to drive hunting behavior towards paramecia in the light, are either blocked by filters or not present within the spectrum of the projector. To clarify this stimulus decision, we added a comment to the Materials and methods section.

b) Stimulus sizes used (5 deg for small, 30 deg for large) seem quite large, and based on the assumption that zebrafish prey (paramecia) are 250 microns in length, which is very large for a paramecium, and it also implies that they are viewed from the side.

As stated in the Materials and methods section, we were using paramecia of the type *Paramecium multimicronucleatum* for our experiments. The size of this type ranges from 180-310 µm (Wichterman, R., 2012. *The Biology of Paramecium*. Springer Science and Business Media. ISBN 9781475703726). 250 µm also is the average size of rotifers, which are frequently used for rearing zebrafish larvae (Aoyama et al., 2015. *A Novel Method for rearing zebrafish by using freshwater rotifers.* Zebrafish. DOI: 10.1089/zeb.2014.1032). Finally, a stimulus size of 5° was successfully used in a prey capture paradigm by Antinucci et al., 2019.

c) Stimuli were red and black, but paramecia are more visible at shorter wavelengths (e.g. Yoshimatsu et al., 2019 bioRxiv – a study "Yoshimatsu et al., 2020" is cited in the Introduction in a different context, but it is missing from the reference list).Rather, equally interesting other visual circuits are probably driven more effectively with the stimuli as used. Would therefore the framing be better attempted around a characterization of size and motion vision in the early visual brain in general, which would be interesting and novel in its own right?Bianco 2011 e.g. showed that black objects "work". Unfortunately the authors insist on linking their findings specifically to paramecia, which simply are not "black". And they are also much smaller that suggested. If they want to link this data to prey capture, they should acknowledge this.

We appreciate the reviewers' concern. We removed our insistence on paramecia, which just happened to be our lab prey, but of course one can imagine dark prey items in this size range, like rotifers (see above).

- added "rotifer" in subsection “Tectal representation of large (close) objects in frontal visual field is required for hunting” and Discussion paragraph six.

- exchanged "paramecium" for "prey" in the Discussion.

It is quite clear that prey detection performance is much higher when the prey appears in front of the fish – the authors themselves have shown this (Mearns et al., 2020), after earlier observations by others (e.g. Bianco, 2011). Specifically, zebrafish show best detection performance when the prey lines up with the acute zone, about 20-30 deg laterally offset, just above the horizon. For this reason e.g. Semmelhack et al., 2014 (also a paper from the authors) put the screen in front of the fish, not to the side. It therefore seems quite clear that the primary prey detection circuit sits in the retina's acute zone. This also makes much more sense – the acute zone has slightly better resolution that the rest of the eye.

It is true that the major objective of larval prey capture is to position the prey in the binocular "strike zone" in front of the fish. However, initiation of prey capture behavior, i.e. approach swimming, is already triggered when a distant, small prey object appears in the peripheral visual field and does not rely on binocular cues (Mearns et al., 2020). Furthermore, zebrafish larvae have their eyes positioned quite laterally and have only a small binocular zone when embedded. Consequently, presenting visual stimuli from the side allowed us to map A-P positional differences for size-selectivity in the tectum, which would not have been as easy had we presented stimuli in front of the fish.